# DiffWind: Physics-Informed Differentiable Modeling of Wind-Driven Object Dynamics

**Yuanhang Lei[1*], Boming Zhao[1*], Zesong Yang[1*], Xingxuan Li[1], Tao Cheng[1],**
**Haocheng Peng[1], Ru Zhang[1], Yang Yang[1,2], Siyuan Huang[2], Yujun Shen[3],**
**Ruizhen Hu[4], Hujun Bao[1], Zhaopeng Cui[1†]**

[*] Equal contribution. [†] Corresponding author.
[1] State Key Laboratory of CAD & CG, Zhejiang University,
[2] State Key Laboratory of General Artificial Intelligence, BIGAI
[3] Ant Group, [4] Shenzhen University

## Abstract

Modeling wind-driven object dynamics from video observations is highly challenging due to the invisibility and spatio-temporal variability of wind, as well as the complex deformations of objects. We present DiffWind, a physics-informed differentiable framework that unifies wind-object interaction modeling, video-based reconstruction, and forward simulation. Specifically, we represent wind as a grid-based physical field and objects as particle systems derived from 3D Gaussian Splatting, with their interaction modeled by the Material Point Method (MPM). To recover wind-driven object dynamics, we introduce a reconstruction framework that jointly optimizes the spatio-temporal wind force field and object motion through differentiable rendering and simulation. To ensure physical validity, we incorporate the Lattice Boltzmann Method (LBM) as a physics-informed constraint, enforcing compliance with fluid dynamics laws. Beyond reconstruction, our method naturally supports forward simulation under novel wind conditions and enable new applications such as wind retargeting. We further introduce *WD-Objects*, a dataset of synthetic and real-world wind-driven scenes. Extensive experiments demonstrate that our method significantly outperforms prior dynamic scene modeling approaches in both reconstruction accuracy and simulation fidelity, opening a new avenue for video-based wind–object interaction modeling. The project page is available at: https://zju3dv.github.io/DiffWind/.

## 1 Introduction

The motion of objects under wind, such as leaves swaying, flags fluttering, or fabrics billowing, is a visually distinctive yet physically complex phenomenon, arising from the interplay between external fluid forces and internal material properties. From visual observations alone, humans can intuit the presence of invisible wind and anticipate how objects would respond under similar conditions. Enabling computational systems to replicate this ability, which involves recovering both the underlying wind field and the resulting object dynamics from visual input, would have broad impact on applications in augmented and virtual reality, visual effects, scientific analysis, and simulation-based editing.

However, this task presents significant challenges: wind is invisible, dynamic, and spatially non-uniform, while object deformations depend on unknown physical parameters such as mass, elasticity, and geometry. Existing methods face significant limitations: Dynamic neural representations, such as Deformable NeRF Du et al. (2021); Chu et al. (2022) and 3D Gaussian Splatting Yang et al. (2024b); Wu et al. (2024), model object appearance and motion over time but capture only visible dynamics, ignoring underlying physical causes like wind fields. Differentiable physics simulators Zhong et al. (2024); Li et al. (2023b); Zhang et al. (2024) can optimize motion parameters, but are restricted to simple, predefined motion patterns such as constant-force projectile motion and cannot handle complex fluid–object interactions. Video-based wind inference methods Zhang et al. (2022); Runia et al. (2020) estimate coarse wind speed or target specific scenarios such as cloth deformation, lacking generality and physical consistency. These limitations motivate the following question: *Can we*

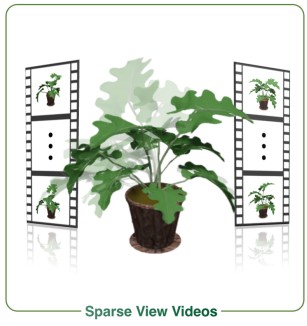 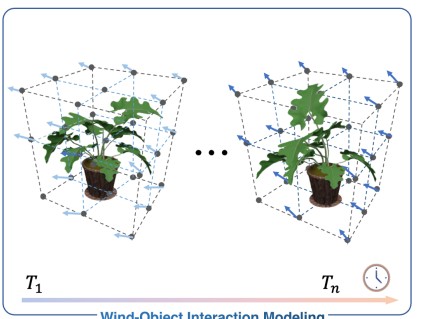 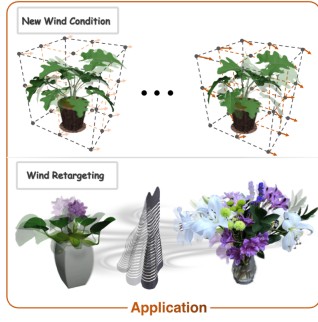

Figure 1: DiffWind is a physics-informed differentiable wind-object interaction framework that models wind and object dynamics separately. This design enables the reconstruction of wind and object motion from sparse-view videos, physically consistent simulation under new wind conditions, and retargeting to novel objects.

*jointly recover visible object dynamics and invisible wind fields from videos input, while ensuring physical consistency and generalization to arbitrary wind conditions?*

To address this challenge, we propose DiffWind, a physics-informed differentiable framework for wind–object interaction. In DiffWind, the invisible wind is modeled as a grid-based physical field, while objects are modeled as deformable particle systems derived from 3D Gaussian Splatting, with their interaction modeled by the Material Point Method (MPM). This design leverages physical intuition: fluids are naturally defined on Eulerian grids, where quantities such as velocity and pressure evolve over space and time, whereas solid or semi-solid objects undergo localized deformation, making Lagrangian particles a more effective representation. By coupling these domains, our formulation supports accurate wind-driven object dynamics reconstruction, physically plausible wind–object interaction simulation, and novel applications such as wind retargeting. (Fig. 1)

A key strength of DiffWind is its differentiable formulation, which enables joint reconstruction of the spatio–temporal wind force field and dynamic object motion from sparse-view RGB videos. Reconstruction is achieved by minimizing the image-space reconstruction error via gradient-based optimization. However, photometric loss alone cannot guarantee physical validity, as wind forces follow complex fluid interactions that often entangle physical properties with scene-specific dynamics, limiting transferability. To address this, we introduce a *physics-informed optimization loss* that enforces compliance with fluid dynamics. Specifically, we use the Lattice Boltzmann Method (LBM) Li et al. (2023a) to provide directional guidance for the wind force field at each time step, constraining its reconstruction to yield physically plausible results.

Our contributions can be summarized as follows:

- We propose a novel differentiable modeling framework DiffWind for wind-object interaction, where the wind is represented as a grid-based physical field and the object is modeled with particle-based deformable geometry. This particle-grid coupling enables physically plausible, 3D-consistent simulations of wind-induced object motion.
- Based on the proposed interaction model, we develop a differentiable inverse reconstruction framework that simultaneously recovers the dynamic motion of visible objects and the invisible wind force field from sparse-view videos.
- We employ the Lattice Boltzmann Method (LBM) as a physics-informed constraint to ensure the wind force field adheres to fluid dynamics laws, and demonstrate applications in forward simulation under novel wind conditions and wind retargeting.
- We construct WD-Objects, a dataset covering both synthetic and real-world wind-driven object scenes. Extensive experiments show that DiffWind substantially outperforms existing methods in reconstruction accuracy, simulation fidelity, and generalization ability, advancing the frontier of video-based wind–object interaction modeling.

## 2  RELATED WORK

**4D Dynamic Scene Reconstruction.**   Dynamic scene reconstruction and free-viewpoint rendering have been widely explored by extending Neural Radiance Field (NeRF) Mildenhall et al. (2020) and

3D Gaussian Splatting (3DGS) Kerbl et al. (2023). NeRF-based approaches model dynamics either by adding temporal inputs Du et al. (2021); Chu et al. (2022) or by learning canonical spaces with deformation fields Pumarola et al. (2021); Park et al. (2021). 3DGS offers fast reconstruction and high-quality rendering, and has been extended to dynamic scenes via Gaussian tracking Luiten et al. (2024); Zeng et al. (2025), canonical-space deformation Yang et al. (2024b); Wu et al. (2024); Zhao et al. (2024), or 4D Gaussian primitives Duan et al. (2024); Wang et al. (2025). But they mainly capture visible object dynamics and struggle to model complex, invisible effects such as wind fields, limiting their applicability to wind-driven object dynamics modeling.

**Wind-Driven Object Dynamics Modeling.** Wind-driven object dynamics modeling involves: (1) simulating wind-induced dynamics, and (2) reconstructing physical parameters from observed motion. Existing simulations use grid-based or particle-based fluid methods to animate specific objects (e.g., grass, trees, cloth) Pirk et al. (2014); Wilson et al. (2014); Lo et al. (2016), but are often category-specific. We instead employ the Material Point Method (MPM) Hu et al. (2018) to model complex deformations of various categories of objects under wind fields. Reconstruction methods estimate wind speed or material properties from monocular videos Yang et al. (2017); Cardona et al. (2019); Runia et al. (2020), but cannot recover full 4D dynamics. In contrast, our method jointly reconstructs temporally varying wind fields and dynamic scenes via differentiable optimization.

**Differentiable Physics-Based System Identification.** System identification is a methodology for building mathematical models of dynamic systems using measurements of the input and output signals of the system. Recent advances in differentiable physical simulation, enabled by frameworks like Taichi Hu et al. (2019; 2020) and Warp Macklin (2022), have facilitated such tasks. Prior works optimize material or geometry parameters for cloth Li et al. (2022), solids Jin et al. (2024), and fluids Li et al. (2024b) using differentiable solvers. Other studies combine differentiable simulation with NeRF Mildenhall et al. (2020) or 3DGS Kerbl et al. (2023) to jointly recover object geometry and physical properties from multi-view observations Li et al. (2023b); Zhong et al. (2024); Cai et al. (2024); Zhang et al. (2024); Cao et al. (2024); Liu et al. (2025), but are restricted to simple, predefined motion settings (e.g., gravity, dragging) and optimize initial velocity at the first frame. Moreover, these approaches focus solely on object motion and do not model the influence of the external environment such as wind fields on the dynamics. In contrast, our method reconstructs the wind force field through a differentiable optimization process, while enabling the modeling of complex and dynamic wind–object interactions.

## 3 METHOD

**Problem Specification:** Given the sparse-view observed videos as input $\boldsymbol{V} = \{\{I_0^i, ..., I_T^i\} | i = 1, ..., N_v\}$ of the wind-object interaction, our goal is to reconstruct the dynamic process in 3D. Our key idea is to optimize the wind force field to match the observed sequences through differentiable simulation and differentiable rendering.

As shown in Fig. 2, we model the wind as a grid field, where each node can contain various physical quantities. The object is represented as a set of particles, each carrying attributes such as appearance, material, and motion. We employ a differentiable Material Point Method (MPM) to model the interaction between the wind and the object. The *force* property of the wind grid field is applied to the background grid of MPM, which drives the motion of object particles and results in object deformation. More details about the interaction modeling are provided in Sec. 3.2. Then, given observed sequences of object motion under wind influence, we perform differentiable optimization of the *force* property in the wind grid field that enables the reconstruction of the dynamic process (Sec. 3.3). Furthermore, we propose a novel physics-informed optimization method that leverages LBM to enforce the force field's compliance with the fundamental laws of fluid dynamics (Sec. 3.4).

### 3.1 PRELIMINARY: PHYSICAL MODELS

**Fluid Mechanics.** Our wind field dynamics are governed by incompressible Navier-Stokes Equations (NSE). We build the fluid simulator by leveraging the Lattice Boltzmann Method (LBM), which solves a discretized form of the Boltzmann equation and recovers the NSE at the macroscopic level. Owing to its local update rules and explicit time integration, LBM is highly amenable to GPU-parallel

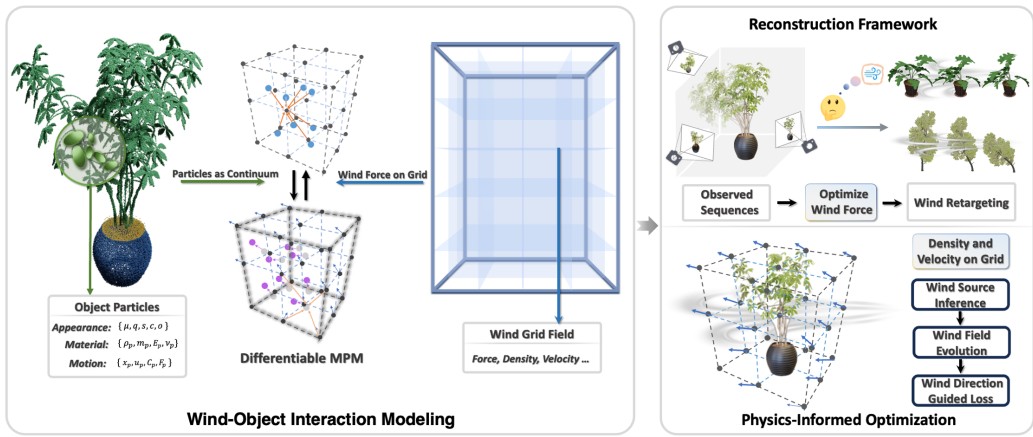

Figure 2: Overview of DiffWind. We propose a novel wind-object interaction modeling approach, where the wind is represented as a grid field and the object is modeled as a set of particles. Based on this modeling approach, we introduce a reconstruction framework for wind–object interaction by optimizing the wind force field. In addition, we employ the Lattice Boltzmann Method (LBM) to generate wind force field direction guidance to enforce compliance with fluid dynamics laws.

computation, offering comparable numerical accuracy to traditional NSE solvers Jasak (2009) while achieving significantly higher computational efficiency. With $t$ discretized into fixed intervals, $\mathbf{x}_w$ discretized on a regular grid and $\mathbf{u}_w$ discretized into fixed unit directions at grid points. Under this discretization, one can solve the NSE by computing the corresponding Lattice Boltzmann Equation (LBE) SHAN et al. (2006):

$$\boldsymbol{f}_i(\mathbf{x}_w + \mathbf{c}_i, t+1) - \boldsymbol{f_i}(\mathbf{x}_w, t) = \Omega_i + \mathbf{F_o}_i, \tag{1}$$

where $\mathbf{c}_i$ is the discrete lattice velocity in the $i$-th direction, $\Omega_i$ is the Bhatnagar-Gross-Krook(BGK) collision model term SHAN et al. (2006) and $\mathbf{F_o}_i$ is the external force. The macroscopic physical quantities of the wind field, such as density $\rho_w$, linear momentum $\rho_w \boldsymbol{u}_w$, and the stress tensor $\boldsymbol{S}_w$ can be calculated as:

$$\rho_w = \sum_{i=0}^{q-1} \boldsymbol{f}_i, \quad \rho_w \boldsymbol{u}_w = \sum_{i=0}^{q-1} \boldsymbol{c}_i \boldsymbol{f}_i + \frac{1}{2}\mathbf{F_o}_i, \quad \rho S_{w,\alpha\beta} = \sum_{i=0}^{q-1} \left( c_i^2 - \frac{1}{3}\delta_{\alpha\beta} \right) \boldsymbol{f}_i, \tag{2}$$

where $\alpha, \beta \in \{x, y, z\}$ refer to the coordinates of the stress tensor $\boldsymbol{S}_w$ and $\delta$ is Kronecker delta. In particular, we use HOME-LBM Li et al. (2023a) as our simulator, which solves the LBE through High-Order Moment Encoding with Hermite polynomial expansion SHAN et al. (2006) to achieve more efficient and accurate fluid simulation.

**Continuum Mechanics.** Our object deformations are governed by continuum mechanics which describes motions by a time-dependent deformation map $\boldsymbol{x}_o = \phi(\boldsymbol{X}_o, t)$ that relates rest material position and deformed material position and the deformation gradient $\boldsymbol{F}(\boldsymbol{X}_o, t) = \nabla_{\boldsymbol{X}_o}\phi(\boldsymbol{X}_o, t)$. The evolution of $\phi$ is governed by:

$$\frac{\partial\rho_o}{\partial t} + \boldsymbol{u}_o \cdot \nabla\rho_o + \rho_o\nabla \cdot \boldsymbol{u}_o = 0, \quad \rho_o(\frac{\partial\boldsymbol{u}_o}{\partial t} + \boldsymbol{u}_o \cdot \nabla\boldsymbol{u}_o) = \nabla \cdot \sigma + \rho_o F_w, \tag{3}$$

where $\rho_o$ is the mass density, $\boldsymbol{u}_o$ is the local material velocity of the object, $\sigma = \frac{1}{J}\mathbf{P}(\boldsymbol{F})\boldsymbol{F}^T$ is the Cauchy stress. These two equations respectively describe the conservation of mass and momentum. In particular, we use the differentiable Material Point Method Hu et al. (2020) to evolve Eq. (3).

## 3.2 WIND-OBJECT INTERACTION MODELING

We represent the object with 3D Gaussians, a particle-based approach suited for high-quality reconstruction and rendering. The wind, being invisible and difficult to represent with discrete particles, is modeled as a continuous grid-based field, where each grid node stores physical quantities such as density, velocity, and force, and its evolution is simulated using the Lattice Boltzmann Method (LBM). To model the interaction between the two representations, we adopt the Material Point

Method (MPM), which exchanges information between particle-based objects and the grid-based wind: objects are treated as moving Lagrangian particles, while wind acts as a force field on an Eulerian grid.

**Object Modeling.** To integrate 3D Gaussians into physical simulations, the representation should minimize rendering artifacts under deformation, accurately capture surfaces for boundary conditions, preserve volume integrity, and support region-wise physical properties. Vanilla 3DGS, however, may produce needle-like artifacts and floaters under large deformations, lacks internal structure, and creates holes in weakly textured regions. We mitigate these issues by adding regularization terms during optimization and densifying invisible points in geometry-deficient areas.

In particular, we define the following loss to optimize the 3D Gaussians:

$$\mathcal{L}_{\text{static}} = \mathcal{L}_{\text{color}} + \lambda_a \mathcal{L}_{\text{aniso}} + \lambda_o \mathcal{L}_{\text{o}}, \tag{4}$$

where $\mathcal{L}_{\text{color}}$ is the color loss between render and training image in vanilla 3DGS; $\mathcal{L}_{\text{aniso}}$ is designed to constrain the anisotropy of the Gaussian kernels:

$$\mathcal{L}_{\text{aniso}} = \frac{1}{|P|} \sum_{i \in P} \max\left( \frac{\max\left(\{s_i^x, s_i^y, s_i^z\}\right)}{\text{median}\left(\{s_i^x, s_i^y, s_i^z\}\right)}, \epsilon \right) - \epsilon, \tag{5}$$

where $\epsilon = 2.0$ is a hyperparameter that controls the level of anisotropy; and $\mathcal{L}_{\text{o}}$ is designed to encourage the 3D Gaussian points to stay close to the object surface by encouraging each Gaussian's opacity to be either near zero or near one:

$$\mathcal{L}_{\text{o}} = \exp\left( -\beta_o \cdot (o_i - 0.5)^2 \right), \tag{6}$$

where $\beta_o = 20$ is a hyperparameter. Additionally, we prune Gaussian points whose opacity falls below a threshold $\tau_o = 0.05$.

Based on the optimized Gaussian kernels, we employ the octree-based voxel filling algorithm from Kaolin Fuji Tsang et al. (2024) to densify additional invisible internal points for improved geometric support. Since 3DGS directly models the entire scene, extracting objects for simulation becomes necessary. Moreover, physical properties related to object deformation cannot be directly obtained from static image observations. Therefore, we train a 3D-consistent feature field by contrastive learning Ying et al. (2024); Yang et al. (2025) to extract 2D segmentation priors for 3D regions segmentation in the scene, and employ a multimodal large language model (MLLM) to construct a physical agent that infers their corresponding physical properties, including material name, density, Poisson's ratio $\nu_p$, and Young's modulus $E$. For more details about 3D regions segmentation and the physical agent, please refer to the Appendix.

**Interaction Modeling.** We employ Material Point Method (MPM) to model the interaction between the object and wind, by binding each Gaussian kernel and densified internal point to a corresponding particle in the MPM simulation. During the *P2G* and *G2P* process in MPM, wind is applied as a force field to the MPM grid to simulate the deformation of objects. Similar to PhysGaussian Xie et al. (2024), we update the covariance matrix of 3DGS as $\Sigma_p(t) = \boldsymbol{F}_p(t)\Sigma_p \boldsymbol{F}_p(t)^T$, $\boldsymbol{F}_p(t)$ is the deformation gradient of each particle $p$ at time step $t$.

### 3.3 RECONSTRUCTION FRAMEWORK FOR WIND-OBJECT INTERACTION

We formulate the simulation process of wind-object interaction for a single step as:

$$(\boldsymbol{x}^{t+1}, \mathbf{v}^{t+1}, \boldsymbol{F}^{t+1}, \boldsymbol{C}^{t+1}) = \mathcal{S}(\boldsymbol{x}^t, \mathbf{v}^t, \boldsymbol{F}^t, \boldsymbol{C}^t, \theta, \mathbf{F}_w, \Delta t), \tag{7}$$

where $(\boldsymbol{x}^t, \mathbf{v}^t, \boldsymbol{F}^t, \boldsymbol{C}^t)$ denote the position, velocity, deformation gradient, and affine momentum of all object particles at time $t$, respectively. Note that the particles include both Gaussian kernels and densified internal points. $\theta$ denotes the collection of the physical properties of all particles, which are obtained during the object modeling step, and $\mathbf{F}_w$ denotes the wind force field.

We can then render the object at each time and each viewpoint by 3D Gaussian Splatting:

$$\hat{I}_t = \mathcal{R}_{\text{render}}(\boldsymbol{x}_{gs}^t, \sigma, \boldsymbol{F}_{gs}^t, \Sigma, c, W), \tag{8}$$

where $\boldsymbol{x}_{gs}^t$ and $\boldsymbol{F}_{gs}^t$ denote the subset of $\boldsymbol{x}^t$ and $\boldsymbol{F}^t$ corresponding to the Gaussian kernels.

Owing to the differentiable nature of Eq. (7) and Eq. (8), we can optimize the invisible wind force field from the visible motion of the object by minimizing the photometric loss between the rendered image and input image as:

$$\mathcal{L}_{\text{render}} = \sum_{i=1}^{N_v} (1 - \lambda)\mathcal{L}_1^i + \lambda\mathcal{L}_{D-SSIM}^i, \tag{9}$$

where $N_v$ denotes the number of viewpoints, $\mathcal{L}_1^i$ and $\mathcal{L}_{D-SSIM}^i$ represent the $L_1$ and structural similarity index measure losses at viewpoint $i$, respectively. We set $\lambda = 0.1$ in our experiments.

Although $\theta$ also receives gradients during backpropagation, jointly optimizing both the wind force field $\mathbf{F}_w$ and the material parameters leads to an ill-posed problem, because different combinations of material stiffness and external forces can produce nearly identical motions. For instance, a small Young's modulus paired with a small force can generate the same motion as a large Young's modulus with a large force. Therefore, we use MLLM-based reasoning to obtain reasonable material priors as initialization (see Sec. 3.2), and treat only $\mathbf{F}_w$ as the optimization target, initializing it from zero for each time-step optimization. We use the results from object modeling as the initial state and optimize the wind force field between two consecutive states. The optimization proceeds sequentially, with each time step starting only after the previous one has been completed. The optimized object state from the previous step serves as the initial condition for the next which mitigates gradient instability that can occur when performing multi-step simulation followed by a single backward pass. By optimizing the wind force field at each time step, we simultaneously achieve the reconstruction of wind-object interaction dynamics.

**Wind Retargeting.**   By accurately reconstruction of wind-object interaction, the proposed framework enables a novel task: wind retargeting. This involves applying the estimated wind force fields to other objects, as illustrated in Fig. 1, which is infeasible for the existing object dynamics modeling methods like Deformable-GS Yang et al. (2024b), which only focus on dynamic scene reconstruction.

### 3.4 Physics-Informed Optimization of Wind Force Field

Although we can optimize the wind force field in each timestep using visual observations, it does not guarantee adherence to physical laws as wind forces are fluid and exhibit complex interactions with the target objects. Therefore, we further exploit a physics-informed optimization loss to guide the force field optimization process. Specifically, we use the Lattice Boltzmann Method (LBM) to solve Eq. (1), thereby generating the guiding direction of the force field.

**Wind Source Inference.**   We start from the posed RGB video and estimate temporally-consistent monocular depth maps using Video Depth Anything Chen et al. (2025). The RGB–depth pairs are provided to a multimodal large language model (MLLM), which infers the wind source direction. Then, the inferred direction serves as an inlet velocity boundary condition for the LBM simulator.

**Wind Field Evolution.**   We use HOME-LBM Li et al. (2023a) to simulate wind field evolution, where the positions of particles serve as the boundary condition for LBM, thereby influencing the wind field dynamics. In particular, we first distinguish solid and fluid nodes in the LBM lattice based on the positions of particles at the current state by converting object particles into occupied voxel grids:

$$\text{Solid}(\mathbf{x}_w) = \begin{cases} 1, & \text{if } \exists\, \boldsymbol{x}_p \text{ s.t. } \forall d \in \{x, y, z\},\ 0 \leq \boldsymbol{x}_{p,d} - \mathbf{x}_w < \Delta x \\ 0, & \text{otherwise} \end{cases} \tag{10}$$

where $\mathbf{x}_w$ is the position of wind grid node, $\boldsymbol{x}_p$ is the position of particle, and $\Delta x$ is the grid spacing. This mapping enables the LBM simulator to capture the effect of the object's instantaneous position on the wind field. At each simulation step, macroscopic wind field quantities are computed from the distribution functions using Eq. (2). Additional details of the algorithm are provided in the Appendix.

**Physics-Informed Loss.**   The wind velocity field from the LBM simulator $\mathbf{D}_{\text{guide}}$ is used to guide the direction of the reconstructed wind force field $\mathbf{D}_{\text{recon}}$:

$$\mathcal{L}_{\text{phys}} = \|\mathbf{D}_{\text{recon}} - (\mathbf{D}_{\text{recon}} \cdot \mathbf{D}_{\text{guide}})\mathbf{D}_{\text{guide}}\|^2, \tag{11}$$

where $\mathcal{L}_{\text{phys}}$ is the physics-informed optimization loss that measures the directional difference between the reconstructed wind force field and the one obtained from the LBM simulator. Here, the wind

Figure 3: Our reconstruction dataset from one camera view, with PSNR values of our dynamic reconstruction shown above. Please use **Adobe Reader/KDE Okular** to see **animations**.

Figure 4: Qualitative results of novel view synthesis for a selected synthetic wind-object interaction scene. We compare our method with Deformable-GS Yang et al. (2024b), Efficient-GS Katsumata et al. (2024), 4D-GS Wu et al. (2024) and GaussianPrediction Zhao et al. (2024). Please use **Adobe Reader/KDE Okular** to see **animations.**

force is governed by the aerodynamic drag equation $\mathbf{F}_w = \frac{1}{2}\rho_w C_D |\boldsymbol{v}|^2 \cdot \frac{\boldsymbol{v}}{|\boldsymbol{v}|}$, where $C_D$ is the drag coefficient and $\boldsymbol{v}$ is the wind velocity, this formulation implies that an accurate estimation of the wind velocity field direction can effectively guide the reconstruction of the wind force field. Through both physics-informed constraints and visual supervision, we ensure the accuracy and continuity of the reconstructed wind force field.

## 4 EXPERIMENTS

In this section, we first introduce the implementation details of our method (Sec. 4.1), followed by evaluations of the capability in reconstructing wind-driven object dynamics (Sec. 4.2). We then conduct ablation studies (Sec. 4.3) to validate key design choices. Finally, we demonstrate the application of our method to forward simulation of wind-driven object dynamics (Sec. 4.4).

### 4.1 IMPLEMENTATION DETAILS

For real-world captured data, we first use Detector-Free SFM He et al. (2024) to compute the pose of each viewpoint. Subsequently, we use EntitySeg Lu et al. (2023) to generate instance-level 2D segmentation masks and Grounded-SAM Ren et al. (2024) to generate part-level 2D segmentation masks. Afterward, we train 3D Gaussians for 60k iterations. For 3D regions segmentation, we train 20k iterations to build 3D consistent feature fields Ying et al. (2024); Yang et al. (2025), and then use GPT-5 for physical property reasoning. In our experiments, we divide the simulation space into a $128^3$ grid, and for real-world scenes, only the segmented foreground objects are used as input to the simulator. All components of our simulator are implemented using Taichi Hu et al. (2019; 2020), and all our experiments are evaluated on a single NVIDIA GeForce RTX 4090 GPU.

### 4.2 EVALUATION ON RECONSTRUCTION

In this section, we first introduce our datasets, then compare our method with SOTA dynamic scene reconstruction approaches for novel view synthesis, and finally show our wind retargeting results.

**Datasets.** As there are no publicly available datasets for modeling wind-driven object dynamics, we introduce both synthetic and real-world datasets, named *WD-Objects*, in this paper for evaluation.

Table 1: Quantitative comparison on the synthetic dataset for the novel view synthesis. Best results are highlighted as first , second , third .

| Method | Dress | | | Flag | | | Pants | | | Sweater | | |
|---|---|---|---|---|---|---|---|---|---|---|---|---|
| | PSNR(↑) | SSIM(↑) | LPIPS (↓) | PSNR(↑) | SSIM(↑) | LPIPS(↓) | PSNR(↑) | SSIM(↑) | LPIPS(↓) | PSNR(↑) | SSIM(↑) | LPIPS(↓) |
| Deformable-GS | 27.32 | .9715 | .0253 | 40.86 | .9939 | .0057 | 42.36 | .9924 | .0149 | 33.10 | .9557 | .0594 |
| Efficient-GS | 22.79 | .9470 | .0481 | 26.72 | .9483 | .0499 | 34.73 | .9782 | .0385 | 26.06 | .9241 | .0920 |
| 4D-GS | 27.64 | .9729 | .0239 | 36.96 | .9868 | .0147 | 41.33 | .9911 | .0184 | 30.14 | .9556 | .0702 |
| GaussianPrediction | 29.00 | .9773 | .0201 | 39.89 | .9930 | .0067 | 42.11 | .9917 | .0168 | 28.64 | .9465 | .0734 |
| Ours | 41.76 | .9986 | .0007 | 41.86 | .9974 | .0009 | 58.38 | .9998 | .0002 | 41.72 | .9951 | .0021 |

| Method | Ficus | | | ShapeNetPlant | | | Alocasia | | | Average | | |
|---|---|---|---|---|---|---|---|---|---|---|---|---|
| | PSNR(↑) | SSIM(↑) | LPIPS(↓) | PSNR(↑) | SSIM(↑) | LPIPS(↓) | PSNR(↑) | SSIM(↑) | LPIPS(↓) | PSNR(↑) | SSIM(↑) | LPIPS(↓) |
| Deformable-GS | 41.72 | .9967 | .0030 | 38.58 | .9946 | .0047 | 32.79 | .9681 | .0364 | 36.68 | .9818 | .0213 |
| Efficient-GS | 22.67 | .8951 | .0906 | 24.63 | .9467 | .0532 | 28.41 | .9463 | .0533 | 26.57 | .9408 | .0608 |
| 4D-GS | 28.80 | .9508 | .0372 | 32.91 | .9865 | .0115 | 28.96 | .9491 | .0386 | 32.39 | .9704 | .0306 |
| GaussianPrediction | 38.81 | .9920 | .0083 | 37.46 | .9935 | .0061 | 34.65 | .9738 | .0292 | 35.79 | .9811 | .0229 |
| Ours | 48.82 | .9995 | .0003 | 52.54 | .9998 | .0001 | 44.94 | .9980 | .0010 | 47.15 | .9983 | .0008 |

Table 2: Quantitative comparison on the real-world dataset for the novel view synthesis. Best results are highlighted as first , second , third .

| Method | POTHOS | | | HAT | | | POMPON | | | TULIIP | | |
|---|---|---|---|---|---|---|---|---|---|---|---|---|
| | PSNR(↑) | SSIM(↑) | LPIPS(↓) | PSNR(↑) | SSIM(↑) | LPIPS(↓) | PSNR(↑) | SSIM(↑) | LPIPS(↓) | PSNR(↑) | SSIM(↑) | LPIPS(↓) |
| Deformable-GS | 24.95 | .9572 | .0362 | 30.55 | .9575 | .0297 | 26.06 | .9604 | .0303 | 20.49 | .9536 | .0439 |
| Efficient-GS | 22.25 | .9368 | .0573 | 29.53 | .9562 | .0442 | 24.02 | .9477 | .0447 | 19.34 | .9432 | .0707 |
| 4D-GS | 24.37 | .9516 | .0427 | 30.32 | .9556 | .0347 | 26.23 | .9614 | .0298 | 20.41 | .9527 | .0462 |
| GaussianPrediction | 24.69 | .9532 | .0401 | 30.36 | .9564 | .0307 | 25.57 | .9568 | .0348 | 19.96 | .9511 | .0463 |
| Ours | 26.18 | .9692 | .0256 | 31.37 | .9585 | .0281 | 26.14 | .9660 | .0267 | 23.53 | .9719 | .0308 |

For the synthetic dataset, we first reconstruct seven object-level 3D Gaussians from various sources, including NeRF Mildenhall et al. (2020), PhysDreamer Zhang et al. (2024), ShapeNet Chang et al. (2015); Ma et al. (2024), and LoopGaussian Li et al. (2024a). We then simulate dynamic scenes of the objects moving with the wind, further details are provided in Sec. 4.4. Next, we select eight orthogonal views to render reference videos, where four viewpoints are used for training and the remaining four for evaluating the quality of novel view synthesis. The evaluation is conducted over 50 simulation time steps. We show the object states from one camera view in Fig. 3. For a more complete visualization of the motion process, please refer to our video supplementary. For the real-world dataset, we record 360-degree surround videos of real-world static scenes by GoPro cameras, each scene includes an object and a background. The objects include a pothos plant, a beanie hat, a pompon flower, and a tulip. Additionally, we capture synchronized videos of these plants being affected by a hairdryer.

**Baselines.** We compare our approach against the current state-of-the-art dynamic scene reconstruction methods: Deformable-GS Yang et al. (2024b), Efficient-GS Katsumata et al. (2024), 4D-GS Wu et al. (2024) and GaussianPrediction Zhao et al. (2024). For a fair comparison, we run their optimization process using our reconstructed static 3D Gaussians as initialization, and apply the same regularization terms as described in Sec. 3.2.

**Comparison on Novel View Synthesis.** The quantitative results are shown in Table 1 and Table 2. We present the PSNR/SSIM/LPIPS (VGG) values for the novel view rendering results to validate the accuracy of our wind force field reconstruction results. We can see that the proposed method significantly outperforms existing methods. A set of qualitative results is shown in Fig. 4, with full results provided in the Appendix, demonstrating that our method produces more realistic novel views than existing approaches, owing to its superior modeling of wind-driven object dynamics.

## 4.3 ABLATION STUDIES

**Effectiveness of Physics-Informed Loss.** We evaluate the effectiveness of physics-informed loss during dynamic reconstruction. The quantitative results in Table 3, show that our full model, by embedding physical directional constraints during training, further significantly enhances rendering quality, resulting in reconstructions that are not only more precise but also more physically plausible.

Table 3: Ablation of effectiveness of physics-informed loss on the synthetic dataset.

| Config. | PSNR | SSIM | LPIPS $\times 10^{-1}$ |
|---|---|---|---|
| $w/o\ \mathcal{L}_{\text{phys}}$ | 51.31 | .9995 | .0030 |
| Full Model | 53.24 | .9997 | .0021 |

Ours                     DynamiCrafter                    CogVideoX

Figure 5: Wind synthesis results at the same viewpoint across different timesteps. DynamiCrafter Xing et al. (2023) fails to maintain temporal coherence, CogVideoX Yang et al. (2024a) produces unrealistic jittering. In contrast, DiffWind generates realistic time-evolving wind-object interactions. Please use **Adobe Reader/KDE Okular** to see **animations.**

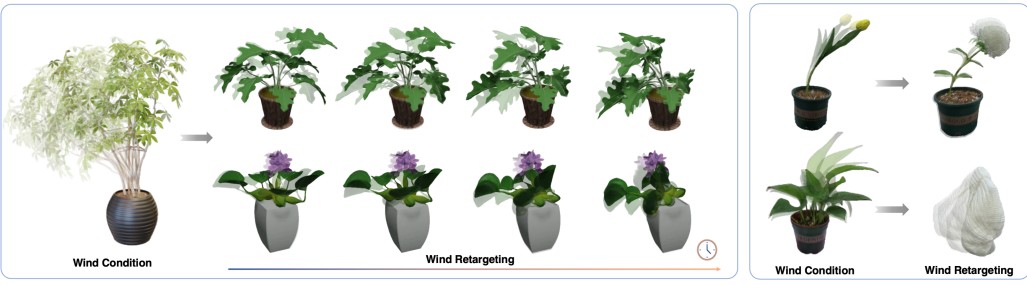

Figure 6: Wind retargeting on synthetic data (left) and real-world data (right).

**Robustness against Physical Property.** We also inspect the robustness of our method to variations in material parameters. To this end, we conduct experiments on the synthetic dataset to analyze how changes in Young's modulus affect reconstruction performance. We predefine three categories of Young's modulus: *soft*, *medium*, and *hard*, with values of $E = 4 \times 10^4$, $4 \times 10^5$, and $4 \times 10^6$, respectively, and also include values inferred using MLLM.

Table 4: Ablation study of robustness against physical property on the synthetic dataset.

| Config. | PSNR | SSIM | LPIPS $\times 10^{-1}$ |
|---|---|---|---|
| Soft | 49.78 | .9993 | .0040 |
| Medium | 49.83 | .9992 | .0044 |
| Hard | 48.64 | .9988 | .0068 |
| MLLM | 51.31 | .9995 | .0030 |

We perform dynamic reconstruction for each case without the physics-informed loss, in order to better assess the influence of physical properties on the reconstruction results. Table 4 shows that using different predefined physical parameters still achieves comparable reconstruction performance, while the values inferred by MLLM yield slightly better results. Hence, we adopt MLLM for more convenient and accurate physical property estimation.

**Necessity of Invisible Internal Points Densification.** We also evaluate the impact of invisible internal points densification, which provides geometric support for the simulation process. The quantitative results in Table 5 indicate that training with densified internal points improves the reconstruction results, justifying its necessity for accurate geometric modeling.

Table 5: Ablation study of necessity of invisible internal points densification on the real-world dataset.

| Config. | PSNR | SSIM | LPIPS $\times 10^{-1}$ |
|---|---|---|---|
| w/o Densification | 27.31 | .9614 | .0292 |
| Full Model | **27.90** | **.9646** | **.0268** |

**Influence of Grid Resolution.** To evaluate how discretization resolution affects reconstruction quality and computational efficiency, we compare grid sizes of $64^3$, $128^3$, and $256^3$. As shown in Table 6 the quantitative results show that reconstruction quality remains stable across different grid resolutions, while $128^3$ provides the best trade-off between fidelity and runtime. Under our default $128^3$ setting,

Table 6: Ablation study of influence of grid resolution on the synthetic dataset.

| Config. | PSNR | SSIM | LPIPS $\times 10^{-1}$ |
|---|---|---|---|
| $64^3$ | 49.08 | .9991 | .0046 |
| $128^3$ | 51.31 | .9995 | .0030 |
| $256^3$ | 51.90 | .9997 | .0024 |

each optimization iteration, including the LBM update, the MPM simulation, the differentiable rendering, and the gradient backpropagation, takes approximately 1.17 seconds on a single NVIDIA RTX 4090 GPU, compared to 1.02 seconds for $64^3$ and 2.02 seconds for $256^3$.

### 4.4 Forward Simulation of Wind-Driven Object Dynamics

**Simulation under Specified Wind Conditions.** Benefiting from our wind–object interaction modeling approach, we enable forward simulation of wind-driven object dynamics, where object deformations are simulated by MPM and wind field evolution is simulated by LBM. To evaluate the visual quality of our wind-object simulations, we compare our method against SOTA and publicly available video generation models, including SVD Blattmann et al. (2023), CogVideoX Yang et al. (2024a), VideoCrafter Chen et al. (2024), and DynamiCrafter Xing et al. (2023). The comparison is performed on four test scenes. The multi-view images of the static test scenes, Vase and Garden, are sourced from NeRF Mildenhall et al. (2020) and Feature-Splatting Qiu et al. (2024), respectively. Additionally, we captured two static scenes ourselves, Cloth and Sock.

We conduct a user study with 32 participants, who rate 20 randomly ordered videos generated by different methods. Both motion realism and the physical realism of wind–object interactions are evaluated on a 5-point Likert scale, from 1 (strongly disagree) to 5 (strongly agree). Results in Table 7 demonstrate the superior visual quality of our simulations, with comparative examples shown in Fig. 5. Additional results are provided in the Appendix and supplementary video.

Table 7: Human evaluation on visual quality.

| Method | Visual Quality | | | | |
|---|---|---|---|---|---|
| | Cloth | Garden | Sock | Vase | Avg |
| SVD | 2.59 | 2.53 | 2.66 | 2.19 | 2.49 |
| CogVideoX | 2.31 | 1.72 | 2.69 | 2.63 | 2.34 |
| VideoCrafter | 2.56 | 2.44 | 2.38 | 1.94 | 2.33 |
| DynamiCrafter | 1.97 | 3.33 | 2.69 | 3.44 | 2.84 |
| Ours | **4.47** | **4.16** | **4.59** | **4.13** | **4.34** |

**Wind Retargeting.** By using the reconstructed wind force field as input for forward simulation, our method enables a new application, i.e., wind retargeting. We present several representative results of wind retargeting in Fig. 1, Fig. 2 and Fig. 6, with additional examples provided in our supplementary video. It can be observed that our method effectively decouples the invisible wind force field from the visible object dynamics, while enabling accurate transfer to previously unseen scenes.

## 5 Conclusion

In this paper, we present a novel framework for modeling wind-object interactions. The proposed framework can reconstruct wind-driven object dynamics, simulate in new wind conditions and perform wind retargeting. To build this framework, we represent wind as a grid-based physical field and model objects using particle-based deformable geometry. By leveraging differentiable physical simulation and rendering, our system achieves backward reconstruction of wind-object interactions. In addition, we employ the LBM as a physics-informed constraint to enforce compliance with fluid dynamics laws. We also present a new dataset for comprehensive evaluation. The experiments demonstrate the superiority of the proposed framework in modeling realistic wind-object dynamics.

**Limitation and Future Work**. The proposed framework currently focuses on modeling object-level dynamics under wind conditions, without accounting for interactions between objects. Robustly handling multi-object collisions is challenging due to discontinuous contact dynamics and complex force interactions, which complicate gradient-based optimization. In addition, MPM is primarily used for simulating continuum objects, however, our framework is simulator-agnostic and can be extended to non-continuum objects by replacing MPM with an appropriate differentiable simulator without changing the overall formulation, other types of simulators such as the spring–mass method or the Finite Element Method (FEM) could be explored to model a wider range of object motion behaviors. We leave these extensions as future work.

### Acknowledgments

This work was supported by the National Key R&D Program of China (Grant No. 2024YFB4505500 & 2024YFB4505501), the NSFC (No. 62441222, No. 62572425 and No. 624B2132), Ant Group, Information Technology Center, and State Key Lab of CAD&CG, Zhejiang University. We also express our gratitude to all the anonymous reviewers for their professional and insightful comments.

## ETHICS STATEMENT

Our study focuses on modeling wind-driven object dynamics using physics-informed differentiable simulations. All experimental evaluations are conducted using synthetic, publicly available, or self-captured datasets, curated to avoid sensitive or private content. We assert that this research has been carried out in accordance with the code of ethics.

## REPRODUCIBILITY STATEMENT

In order to support reproducibility and verification, we present implementation and evaluation details in our paper, and will release the associated source code upon acceptance.

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

APPENDIX

We highly encourage watching the **supplementary video** where our results are best illustrated.

## A    BACKGROUND

### A.1    3D GAUSSIAN REPRESENTATION

3D Gaussian Splatting (3DGS) Kerbl et al. (2023) employs a substantial number of explicit 3D Gaussians to represent a static 3D scene. Each 3D Gaussian $G$ is defined by a full covariance matrix $\Sigma$ and a center location $\mu$:

$$G(x) = e^{-\frac{1}{2}(x-\mu)^T \Sigma^{-1}(x-\mu)}. \tag{12}$$

For differentiable rendering optimization, 3DGS decomposes $\Sigma$ into scaling matrix $S$ and rotation matrix $R$: $\Sigma = RSS^T R^T$, where $S$ and $R$ are stored by a 3D vector $s$ and a quaternion $q$ respectively. To project these 3D Gaussians to 2D image, given a viewing transformation $W$, we obtain the 2D covariance matrix $\Sigma'$ and 2D center location $\mu'$ as:

$$\Sigma' = JW\Sigma W^T J^T, \quad \mu' = JW\mu, \tag{13}$$

where $J$ is the Jacobian of the affine approximation of the projective transformation. Then we can use the neural point-based $\alpha$-blending to render the color $C$ of each pixel with $N$ ordered 3D Gaussians:

$$C = \sum_{i \in N} T_i c_i \alpha_i, \tag{14}$$

where $T_i, \alpha_i$ are calculated as:

$$T_i = \prod_{j=1}^{i-1}(1 - \alpha_j), \quad \alpha_i = o_i e^{-\frac{1}{2}(x-\mu')^T \Sigma'^{-1}(x-\mu')}. \tag{15}$$

Here, $o_i$ is the opacity of the 3D Gaussian. Therefore, the 3D scene can be represented by the parameter set $P$ of 3D Gaussians, where $P = \{G_i : \mu_i, q_i, s_i, c_i, o_i\}$.

### A.2    FLUID MECHANICS

**Navier-Stokes Equations.**    Navier-Stokes equations are well-known fundamental equations that describe fluid mechanics:

$$\frac{\partial \boldsymbol{u}_w}{\partial t} + \boldsymbol{u}_w \cdot \nabla \boldsymbol{u}_w = -\frac{1}{\rho_w}\nabla p + \nu \nabla \cdot \nabla \boldsymbol{u}_w + \mathbf{F_o},$$
$$\nabla \cdot \boldsymbol{u}_w = 0, \tag{16}$$

where $\boldsymbol{u}_w$ is the wind velocity field, $\rho_w$ is the density, $p$ is the pressure, $\nu$ is the kinematic viscosity and $\mathbf{F}_o$ is the external force, which represents the influence of objects interacting with the wind field in our work. The first equation describes the momentum conservation, while the second equation enforces the incompressibility constraint.

**Continuous Boltzmann Equation.**    The Boltzmann equation was proven to recover Navier-Stokes equation macroscopically SHAN et al. (2006). It describes fluid dynamics through the time evolution of a mesoscopic distribution function $\boldsymbol{f}(\mathbf{u}_w, \mathbf{x}_w, t)$:

$$\frac{\partial \boldsymbol{f}}{\partial t} + \mathbf{u}_w \cdot \nabla \boldsymbol{f} = \Omega(\boldsymbol{f}) + \mathbf{F_o} \cdot \nabla_{\mathbf{u}_w} \boldsymbol{f}, \tag{17}$$

where $\mathbf{F_o}$ represents external forces, $\boldsymbol{f}$ represents the probability of a particle being at position $\mathbf{x}$ at time $t$ with velocity $\mathbf{u}$ and $\Omega$ is the collision term to relax distribution function towards the equilibrium state, typically using the Bhatnagar-Gross-Krook(BGK) model SHAN et al. (2006). By integrating the microscopic properties, the macroscopic physical quantities of the wind field, such as density $\rho_w$, linear momentum $\rho_w \boldsymbol{u}_w$, and the stress tensor $\boldsymbol{S}_w$ can be calculated:

$$\rho_w = \int \boldsymbol{f} \, d\mathbf{u}_w, \quad \rho_w \boldsymbol{u}_w = \int \mathbf{u}_w \boldsymbol{f} \, d\mathbf{u}_w,$$
$$\rho_w \boldsymbol{S}_{w,\alpha\beta} = \int \left( \mathbf{u}_w^2 - \frac{1}{3}\delta_{\alpha\beta} \right) \boldsymbol{f} \, d\mathbf{u}_w, \tag{18}$$

where $\alpha, \beta \in \{x, y, z\}$ refer to the coordinates of the stress tensor $\boldsymbol{S}_w$ and $\delta$ is Kronecker delta.

We note that all numerical solvers for fluid dynamics, including the Lattice Boltzmann Method (LBM) adopted in our framework, face inherent limitations under highly turbulent flows or when the local Reynolds number exceeds the stability range of the scheme. In such regimes, solvers may fail to fully resolve small-scale vortical structures or highly chaotic flow patterns. In our reconstruction pipeline, LBM is used as a physics-informed constraint to regularize wind estimation toward physically plausible behaviors. Consequently, our method can reliably reconstruct moderately complex and unsteady wind fields, but consistent with the intrinsic limitations of the underlying solver, it does not capture the full spectrum of fine-grained turbulent structures.

## A.3 CONTINUUM MECHANICS

The object deformations are governed by continuum mechanics. In this section, we introduce the relevant background.

**Constitutive Models.** Constitutive models describe the material responses to different mechanical loading conditions, which provide the stress–strain relations to formulate the governing equations. We use the Neo-Hookean constitutive model to capture the non-linearity of the object's response to the wind force field:

$$\psi(\boldsymbol{F}) = \frac{\mu}{2} \sum_i \left[ (\boldsymbol{F}^T \boldsymbol{F})_{ii} - 1 \right] - \mu \log(J) + \frac{\lambda}{2} \log^2(J),$$

$$\mathbf{P}(\boldsymbol{F}) = \frac{\partial \psi}{\partial \boldsymbol{F}} = \mu(\boldsymbol{F} - \boldsymbol{F}^T) + \lambda \log(J) \boldsymbol{F}^{-T}, \tag{19}$$

where $\psi$ is the hyperelastic energy density function, $\mathbf{P}$ is the first Piola-Kirchhoff stress, $\boldsymbol{F}$ is the deformation gradient that encodes local transformations including stretch, rotation, and shearBonet & Wood (1997) and $J = det(\boldsymbol{F})$. The Lame parameters $\mu$ and $\lambda$ are related to Young's modulus $E$ and Poisson's ratio $\nu_p$:

$$\mu = \frac{E}{2(1 + \nu_p)}, \quad \lambda = \frac{E\nu_p}{(1 + \nu_p)(1 - 2\nu_p)}. \tag{20}$$

**Material Point Method.** Material Point Method (MPM)Stomakhin et al. (2013); Hu et al. (2018) solves the governing equations by transferring physical properties between Lagrangian particles and Eulerian grid. In particular, we use MLS-MPMHu et al. (2018) as our simulator. Each particle's state is described by a five-tuple $(\boldsymbol{x}_p, \boldsymbol{m}_p, \boldsymbol{u}_p, \boldsymbol{C}_p, \boldsymbol{F}_p)$: $\boldsymbol{x}_p$ is the particle position, $\boldsymbol{m}_p$ is the particle mass, $\boldsymbol{u}_p$ is the particle velocity, $\boldsymbol{C}_p$ is the affine momentumJiang et al. (2015) on particle, $\boldsymbol{F}_p$ is the elastic deformation gradient tracked on the particle. MPM operates in a particle-to-grid (P2G) and grid-to-particle (G2P) transfer loop to simulate the object dynamics. In the P2G stage, we transfer mass and momentum from particles to the grid as:

$$m_i^t = \sum_p w_{ip}^t m_p, \quad m_i^t \boldsymbol{u}_i^t = \sum_p w_{ip}^t m_p \left( \boldsymbol{u}_p^t + \boldsymbol{C}_p^t \left( \boldsymbol{x}_i - \boldsymbol{x}_p^t \right) \right). \tag{21}$$

In the G2P stage, we transfer velocities back to particles and update particle states as:

$$\boldsymbol{u}_p^{t+1} = \sum_i \boldsymbol{u}_i^{t+1} w_{ip}^t, \quad \boldsymbol{x}_p^{t+1} = \boldsymbol{x}_p^t + \Delta t \boldsymbol{u}_p^{t+1}, \tag{22}$$

where $i$ represents the property on the Eulerian grid, $n$ represents the property at time $t_n$ and $w_{ip}^n$ is the B-spline kernel weight at $\boldsymbol{x}_p^n$. The deformation gradient is updated by MLS approximation as:

$$\boldsymbol{F}_p^{t+1} = (\boldsymbol{I} + \Delta t \boldsymbol{C}_p^t) \boldsymbol{F}_p^t. \tag{23}$$

For more details about the simulation algorithm of MLS-MPMHu et al. (2018), we refer the reader to its original publication.

## B MORE DETAILS ON 3D REGIONS SEGMENTATION

A number of existing methods support 3D Gaussians segmentation. Specifically, we employ a contrastive learning strategy to lift 2D segmentation results into 3D Gaussians, following Om-niSeg3DYing et al. (2024). In particular, we use EntitySegLu et al. (2023) to generate instance-level

2D segmentation masks and Grounded-SAM Ren et al. (2024) to generate part-level 2D segmentation masks, where we design prompts for GPT-4o to identify the names of different parts in the scene as input to Grounded-SAM. For each 2D region, we blend its mask with a canonical view reference image, and concatenate it with the reference image as a prompt image, then we guide the physical agent to identify the region in the marked area and explain all possible material types with physical properties, including Poisson's ratio $\nu_p$ and Young's modulus $E$. Next, we associate each Gaussian kernel with an optimizable feature vector $h_g \in \mathbb{R}^{16}$, for the features with region-wise labels $\mathbf{L} = \{l_j^i\}$ in 2D level, we maximize the similarity for features with the same region and low similarity between different regions. We design the following loss for $\mathbf{L}$ as:

$$\mathcal{L}_{CC}(\mathbf{L}) = -\frac{1}{N_r} \sum_{i=1}^{N_r} \sum_{j=1}^{n_i} \log \frac{\exp(l_j^i \cdot \bar{l}^i / \phi_i)}{\sum_{k=1}^{N_r} \exp(l_j^i \cdot \bar{l}^k / \phi_k)}, \qquad (24)$$

where $N_r$ is the number of regions involved in $\mathbf{L}$, $n_i = |\{l^i\}|$, $l_j^i$ is the render feature with point index $j$ and region id $i$, $\bar{l}^i$ is the mean value for $l_j^i$, and $\phi_i = \frac{\sum_{j=1}^{n_i} \|l_j^i - \bar{l}^i\|_2}{n_i \log(n_i + \alpha)}$ is the region temperature, where $\alpha = 10$ is a smoothing parameter. After training the feature field, we apply HDBSCAN clustering algorithmMcInnes et al. (2017) to automatically classify different regions without requiring a predefined number of clusters, we then compute the convex hull of each region to categorize the densified invisible points and Gaussian points that may not have been classified by HDBSCAN and we employ KNN to assign any remaining unclassified points based on their proximity to existing clusters. Finally, we assign physical properties to each 3D region based on the correspondence between 2D and 3D segmentations.

## C   More Details on Densify Invisible Points

Although PhysGaussian proposes using a 3D opacity field from 3D Gaussians for internal filling, in practice, this method often fails to achieve satisfactory results. Moreover, it requires the 3D Gaussian point cloud to be densely distributed and is not suitable for hole completion. Therefore, we adopt the octree-based voxel filling algorithm provided by the Kaolin library to densify invisible points for improved geometric support. The algorithm mainly consists of the following three steps:

**(1) Voxelization of 3D Gaussians via Octree Construction.** The set of 3D Gaussians is first converted into a voxel representation using a hierarchical algorithm built upon Kaolin's Structured Point Cloud (SPC), which functions as an octree. The axis-aligned bounding box (AABB) of all Gaussians is enclosed within a cubical root node, which is recursively subdivided in an 8-way manner. For each sub-node, a list of overlapping Gaussian IDs is maintained. Only the nodes that contain relevant Gaussian density are further subdivided and checked for overlaps. This process continues until a specified octree resolution level is reached. The frontier nodes of the octree represent a voxelized shell of the 3D Gaussians. Optionally, nodes whose accumulated opacity values fall below a predefined opacity_threshold are culled to remove low-density regions. As a result, this step yields a voxel shell approximation of the Gaussian surface, without including inner volume content.

**(2) Volume Filling via Multi-View Depth Fusion.** To convert the voxelized shell into a volumetric solid, the algorithm performs space carving based on rendered depth maps. Specifically, the SPC is ray-traced from an icosahedral set of camera viewpoints to generate depth maps. These depth maps are then fused into a second sparse SPC, where each node of the octree maintains an occupancy state: *empty*, *occupied*, or *unseen*. The occupancy status of each voxel in a regular 3D grid is determined by querying this SPC. Finally, the union of all *occupied* and *unseen* voxels yields a volumetric representation that better captures the object's solid geometry.

**(3) Dense Point Sampling.** After volume filling, the 3D Gaussians are now represented as dense voxels that include the object's interior volume. A point is sampled at the center of each occupied voxel to construct a dense point cloud. Optionally, a small random perturbation is added to each point. This perturbation is constrained to be small enough to ensure the point remains within the bounds of its voxel. By the end of this step, each voxel contains at most one sampled point, resulting in a uniformly distributed and volume-complete point cloud.

# D    MORE DETAILS ON HOME-LBM ALGORITHM

The post-collision distribution function in HOME-LBM is given by:

$$
\begin{aligned}
f_i = \rho w_i \Bigg[ & 1 + \frac{\boldsymbol{c}_i \cdot \boldsymbol{u}}{c_s^2} + \frac{\boldsymbol{H}^{[2]}(\boldsymbol{c}_i) : \boldsymbol{S}}{2c_s^4} \\
& + \frac{1}{2c_s^6} \Big( H_{xxy}^{[3]}(\boldsymbol{c}_i)(S_{xx}u_y + 2S_{xy}u_x - 2u_x u_x u_y) \\
& + H_{yyy}^{[3]}(\boldsymbol{c}_i)(S_{yy}u_x + 2S_{xy}u_y - 2u_x u_y u_y) \\
& + H_{xxz}^{[3]}(\boldsymbol{c}_i)(S_{xx}u_z + 2S_{xz}u_x - 2u_x u_x u_z) \\
& + H_{zzz}^{[3]}(\boldsymbol{c}_i)(S_{zz}u_x + 2S_{xz}u_z - 2u_x u_z u_z) \\
& + H_{yzz}^{[3]}(\boldsymbol{c}_i)(S_{zz}u_y + 2S_{yz}u_z - 2u_y u_z u_z) \\
& + H_{yyz}^{[3]}(\boldsymbol{c}_i)(S_{yy}u_z + 2S_{yz}u_z - 2u_y u_y u_z) \\
& + H_{xyz}^{[3]}(\boldsymbol{c}_i)(S_{xx}u_y + S_{yz}u_x + S_{xy}u_z - 2u_x u_y u_z) \Big) \Bigg],
\end{aligned}
\tag{25}
$$

where $\boldsymbol{H}$ is the Hermite polynomials, $S$ is the stress tensor and $u$ is the velocity. This post-collision distribution function enables the computation of the microscopic distribution function in the LBM, which can then be integrated into the standard LBM simulation pipeline.

To address the low accuracy problem caused by the BGK collision model in the classical LBM algorithm, HOME-LBM introduces a high-order collision model based on Hermite expansion after the streaming step the update process is given by:

$$
\begin{aligned}
\rho(\boldsymbol{x}, t+1) &= \rho^*, \\
u_\alpha(\boldsymbol{x}, t+1) &= u_\alpha^* + \frac{1}{2\rho^*} F_\alpha, \\
S_{xy}(\boldsymbol{x}, t+1) &= \left(1 - \frac{1}{\tau}\right) S_{xy}^* + \frac{1}{\tau} u_x^* u_y^* + \frac{2\tau - 1}{2\tau\rho^*} \left(F_x u_y^* + F_y u_x^*\right), \\
S_{xx}(\boldsymbol{x}, t+1) &= \frac{\tau - 1}{3\tau} \left(2S_{xx}^* - S_{yy}^* - S_{zz}^*\right) + \frac{1}{3} \left(u_x^{*2} + u_y^{*2} + u_z^{*2}\right) \\
&\quad + \frac{1}{\rho^*} F_x u_x^* + \frac{\tau - 1}{3\tau\rho^*} \left(2F_x u_x^* - F_y u_y^* - F_z u_z^*\right), \\
S_{yy}(\boldsymbol{x}, t+1) &= \frac{\tau - 1}{3\tau} \left(2S_{yy}^* - S_{xx}^* - S_{zz}^*\right) + \frac{1}{3} \left(u_x^{*2} + u_y^{*2} + u_z^{*2}\right) \\
&\quad + \frac{1}{\rho^*} F_y u_y^* + \frac{\tau - 1}{3\tau\rho^*} \left(2F_y u_y^* - F_x u_x^* - F_z u_z^*\right), \\
S_{zz}(\boldsymbol{x}, t+1) &= \frac{\tau - 1}{3\tau} \left(2S_{zz}^* - S_{xx}^* - S_{yy}^*\right) + \frac{1}{3} \left(u_x^{*2} + u_y^{*2} + u_z^{*2}\right) \\
&\quad + \frac{1}{\rho^*} F_z u_z^* + \frac{\tau - 1}{3\tau\rho^*} \left(2F_z u_z^* - F_x u_x^* - F_y u_y^*\right).
\end{aligned}
\tag{26}
$$

Here, the superscript * denotes the values obtained after the streaming step and $F$ denotes the force term derived after the streaming step. For a more detailed mathematical derivation, please refer to the original HOME-LBM paper Li et al. (2023a).

# E    MORE EXPERIMENTAL RESULTS

## E.1    MORE SIMULATION RESULTS

Table 8: Detailed time statistics for forward simulation. #OBG denotes the number of object particles, and #BG indicates the number of background Gaussian particles. "LBM Time", "MPM Time", and "Render Time" represent the average computation time per timestep for LBM simulation, MPM simulation, and 3D Gaussian Splatting rendering, respectively.

| Scene. | #OBG | #BG | LBM Time($\times 10^{-2}$s) | MPM Time(s) | Render Time($\times 10^{-2}$s) |
|---|---|---|---|---|---|
| Cloth | 125223 | 527159 | 0.702 | 0.919 | 0.598 |
| Garden | 52505 | 4484605 | 0.710 | 0.830 | 3.994 |
| Sock | 131794 | 630068 | 0.709 | 0.874 | 0.669 |
| Vase | 145868 | 1944064 | 0.720 | 0.947 | 1.890 |

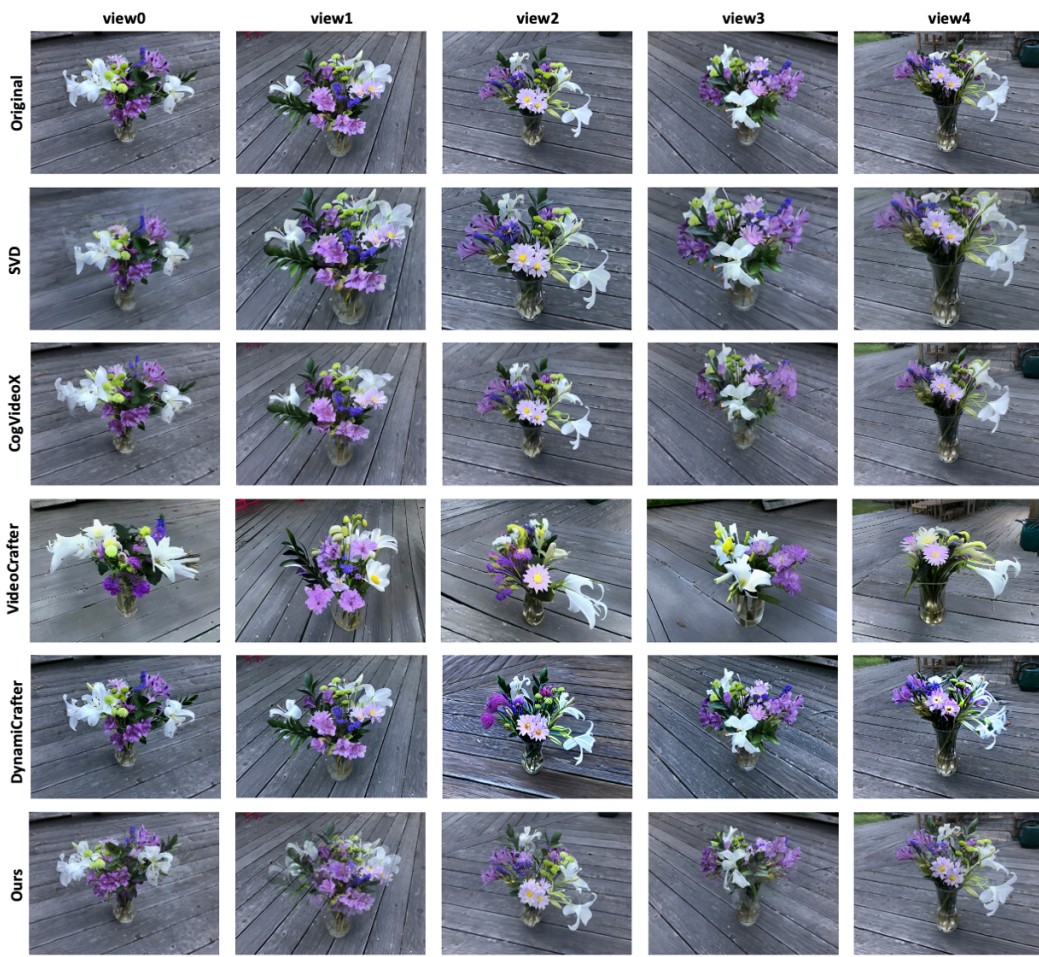

Figure 7: Wind synthesis results at the same timestep from multiple views. SVD Blattmann et al. (2023) exhibits uncontrolled camera movements, VideoCrafter Chen et al. (2024) and Dynami-Crafter Xing et al. (2023) fail to maintain 3D consistency in object geometry and generate only subtle motion, while CogVideoX Yang et al. (2024a) fails to produce geometrically consistent object motion across views. In contrast, DiffWind maintains 3D consistency and produces realistic wind-object interaction.

Fig. 7 and Fig. 9 compare our results against various diffusion models, demonstrating that our method produces more physically plausible and 3D-consistent motion. The time performance of our forward simulation is reported in Table 8. Please refer to our video supplementary material for more results.

### E.2 MORE RECONSTRUCTION RESULTS

**Baselines.** We compare our approach against the current state-of-the-art dynamic scene reconstruction methods: Deformable-GS Yang et al. (2024b), Efficient-GS Katsumata et al. (2024), 4D-GS Wu et al. (2024) and GaussianPrediction Zhao et al. (2024).

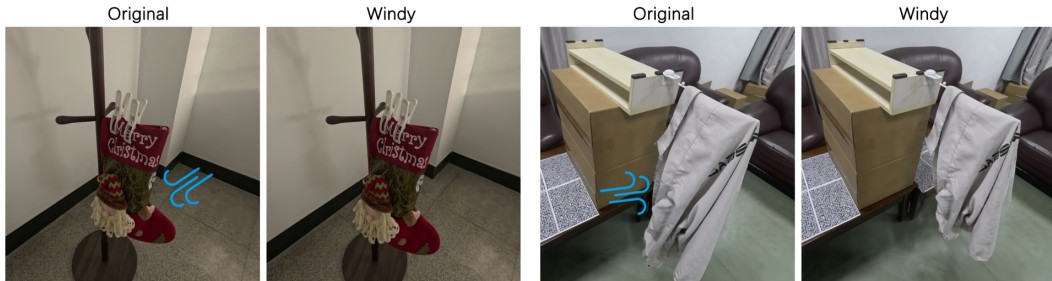

Figure 8: DiffWind is capable of simulating realistic wind-object interactions in static scenes reconstructed from real-world captured multi-view images.

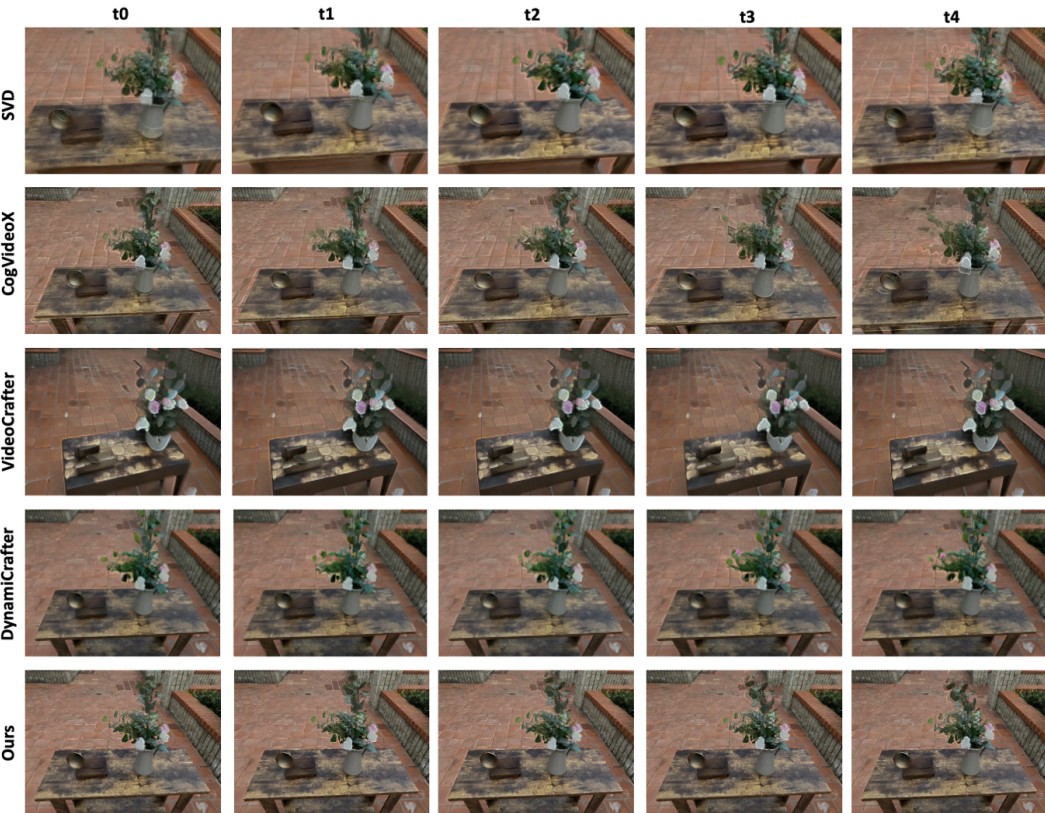

Figure 9: Wind synthesis results at the same viewpoint across different timesteps. SVD Blattmann et al. (2023) suffers from noticeable camera shake, CogVideoX Yang et al. (2024a) produces unrealistic jittering, VideoCrafter Chen et al. (2024) distorts the original object structure and DynamiCrafter Xing et al. (2023) fails to maintain temporal coherence, resulting in implausible motion artifacts. In contrast, DiffWind generates realistic time-evolving wind-object interactions.

Deformable-GS reconstructs dynamic scenes by introducing an additional MLP-based deformation field, while Efficient-GS reconstructs dynamic scenes by learning trajectory functions that govern the motion of Gaussian kernels over time. Similar to Deformable-GS, 4D-GS models the deformation field via HexPlane representations, and GaussianPrediction associates each Gaussian point with an additional motion feature as an extra input to the deformation field. For a fair comparison, we run their optimization process using our reconstructed static 3D Gaussians as initialization, and apply the same regularization terms as described in Sec. 3.2.

**Real-World Data.** We also capture real-world datasets for evaluation. Specifically, we record 360-degree surround videos of real-world static scenes by GoPro cameras. Each scene includes an object and a background. The objects include a pothos plant, a beanie hat, a pompon flower, and a tulip. Additionally, we capture synchronized videos of these plants being affected by a hairdryer. We report the quantitative comparison results in Table. 2 and show the qualitative results in Fig. 11.

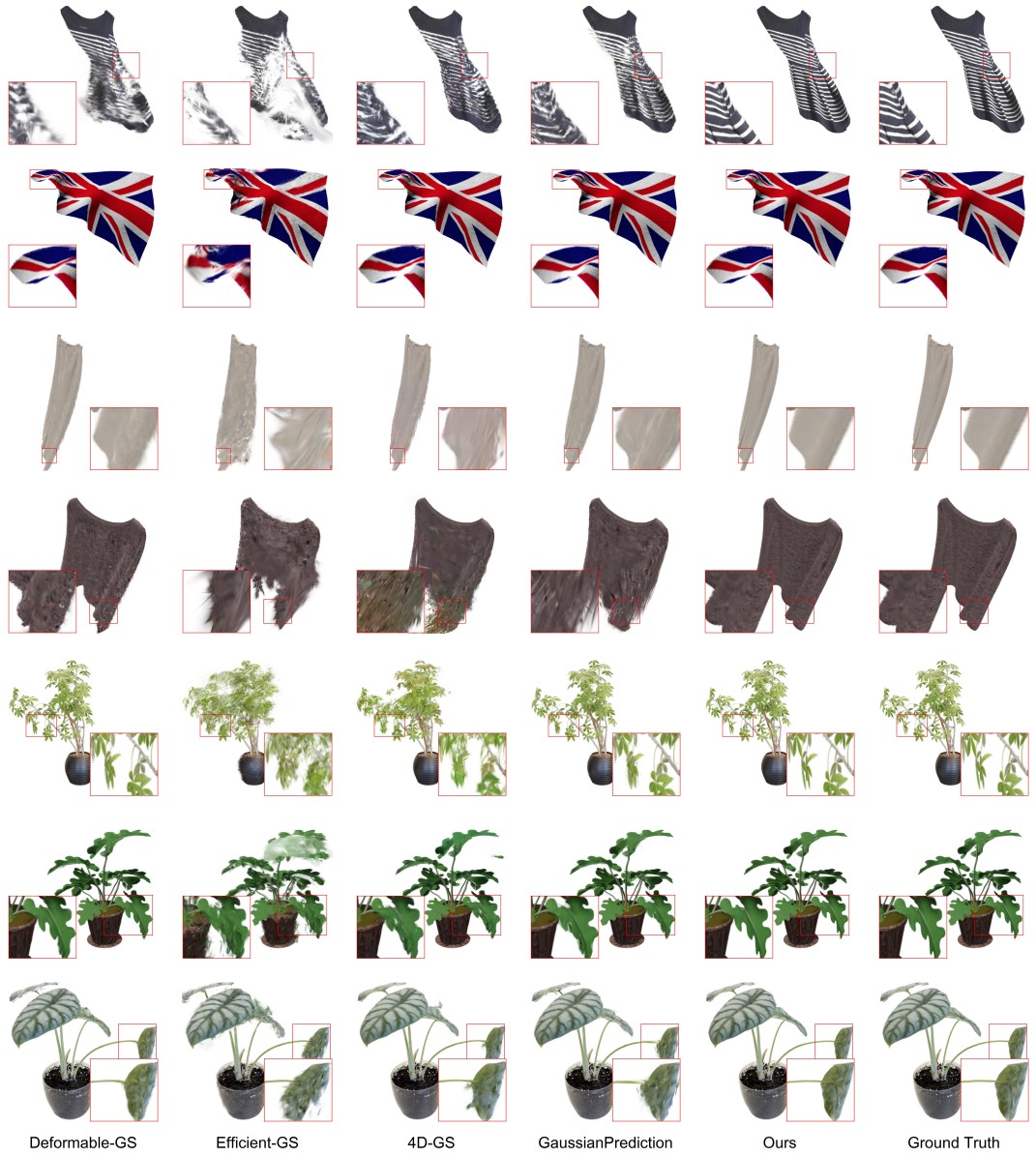

Deformable-GS    Efficient-GS    4D-GS    GaussianPrediction    Ours    Ground Truth

Figure 10: Qualitative results of novel view synthesis on synthetic wind-object interaction scenes. We compare our methods with Deformable-GSYang et al. (2024b), Efficient-GSKatsumata et al. (2024), 4D-GSWu et al. (2024) and GaussianPredictionZhao et al. (2024).

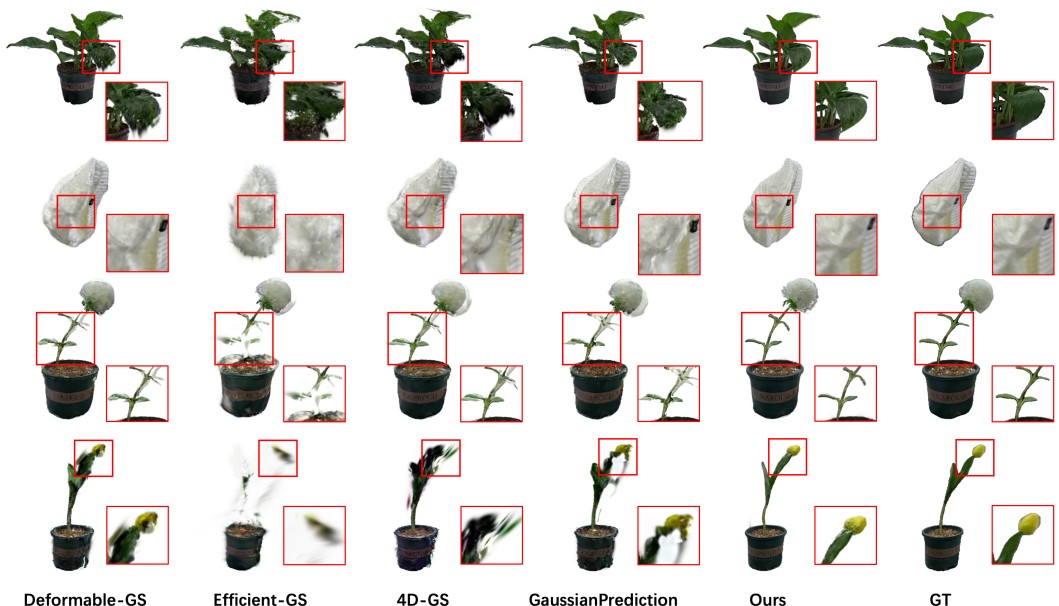

Deformable-GS    Efficient-GS    4D-GS    GaussianPrediction    Ours    GT

Figure 11: Qualitative results of novel view synthesis on real-world wind-object interaction scenes. We compare our method with Deformable-GSYang et al. (2024b), Efficient-GSKatsumata et al. (2024), 4D-GSWu et al. (2024) and GaussianPredictionZhao et al. (2024).

Table 9: Evaluations of reconstructed wind force fields on the synthetic dataset.

| Metrics | Flag | Pants | Sweater | Dress | Ficus | ShapeNet | Alocasia | Average |
|---------|------|-------|---------|-------|-------|----------|----------|---------|
| CosSim | 0.9489 | 0.9585 | 0.9474 | 0.8390 | 0.8695 | 0.9883 | 0.9229 | 0.9249 |
| NMSE | 0.0341 | 0.0276 | 0.0350 | 0.1074 | 0.0870 | 0.0078 | 0.0514 | 0.0500 |

**Ablation on Invisible Point Densification.** We inspect the effectiveness of densifying invisible points and conduct the experiments on the real-world dataset. In Fig. 12, the left side shows the original locations of the 3D Gaussians, while the right side shows the complete point set after densification.

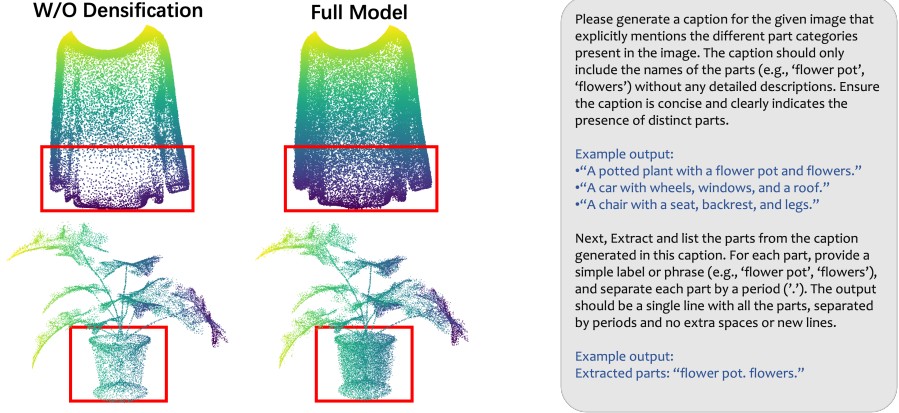

W/O Densification    Full Model

Please generate a caption for the given image that explicitly mentions the different part categories present in the image. The caption should only include the names of the parts (e.g., 'flower pot', 'flowers') without any detailed descriptions. Ensure the caption is concise and clearly indicates the presence of distinct parts.

Example output:
•"A potted plant with a flower pot and flowers."
•"A car with wheels, windows, and a roof."
•"A chair with a seat, backrest, and legs."

Next, Extract and list the parts from the caption generated in this caption. For each part, provide a simple label or phrase (e.g., 'flower pot', 'flowers'), and separate each part by a period ('.'). The output should be a single line with all the parts, separated by periods and no extra spaces or new lines.

Example output:
Extracted parts: "flower pot. flowers."

Figure 12: Visualization of object particle positions before and after invisible points densification.

Figure 13: Prompt for 3D Segmentation.

**Evaluations of Reconstructed Wind Force Fields.** We assess the fidelity of the reconstructed wind force field $\mathbf{F}_w$ under controlled synthetic settings, as shown in Table 9. Let $\mathbf{F}_w^{\text{gt}}$ and $\mathbf{F}_w^{\text{rec}}$ denote the ground-truth and reconstructed wind force fields. We normalize the force vector at each grid location to unit length, and denote the normalized vectors as $\hat{\mathbf{F}}^{\text{gt}}$ and $\hat{\mathbf{F}}^{\text{rec}}$. For directional consistency, we report the average cosine similarity over all time steps $\text{CosSim} = \frac{1}{N}\sum_{i=1}^{N}\hat{\mathbf{F}}_i^{\text{gt}\top}\hat{\mathbf{F}}_i^{\text{rec}}$,

with range $[-1, 1]$, where higher values indicate stronger alignment between reconstructed and ground-truth directions. Additionally, we report the normalized mean squared error over all time steps $\text{NMSE} = \frac{1}{N} \sum_{i=1}^{N} \|\hat{\mathbf{F}}_i^{\text{gt}} - \hat{\mathbf{F}}_i^{\text{rec}}\|^2$, with range $[0, 4]$, where lower values indicate better reconstruction. These results show that the recovered wind force fields are directionally coherent and maintain consistent relative magnitudes with respect to the ground truth on the synthetic dataset.

**Evaluations of Other Baselines.** Note that there are existing methods that also employ differentiable physical simulation for physics parameter estimation from video observation, including PhysDreamer Zhang et al. (2024) and PhysFlow Liu et al. (2025). However, both of them only optimize the initial motion velocities at first frame and Young's modulus for simple and predefined motion patterns, which cannot fully characterize the complex dynamics provided in the input video, especially for windy object reconstruction, as shown in Fig. 14. Therefore, the results obtained by these methods are generally worse than those of the dynamic scene reconstruction methods we listed above for novel view synthesis. For example, for the Ficus example in our dataset, the PSNR/SSIM/LPIPS metrics for PhysDreamer and PhysFlow are 18.22/0.8546/0.0948 and

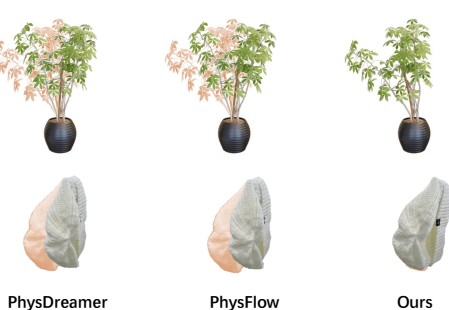

PhysDreamer    PhysFlow    Ours

Figure 14: PhysDreamer and PhysFlow fail to reconstruct dynamics for objects with complex wind-induced motion, with the GT motion (red) shown underneath for reference, while our method faithfully captures the motion.

18.11/0.8529/0.0965, respectively, which are consistently worse than other methods. Hence, we focus on the comparison of our method to those SOTA dynamic scene reconstruction methods for the novel view synthesis task.

### E.3 COMPUTATIONAL COST AND SCALABILITY

All experiments in our paper can be executed on a single NVIDIA RTX 4090 GPU (24GB). Both the LBM and MPM simulators are implemented using the Taichi GPU framework, enabling high-performance GPU execution of the entire physics pipeline. For forward simulation, DiffWind requires less than 1 second per frame for the complete LBM wind update, MPM object simulation, and differentiable rendering, demonstrating higher efficiency than existing video generation baselines, for example, VideoCrafter requires $\sim 81.68$s to generate 16 frames, DynamiCrafter $\sim 151.47$s for 16 frames, and CogVideoX $\sim 292.37$s for 50 frames. For reconstruction tasks, each optimization iteration includes the LBM update, MPM simulation, differentiable rendering, and gradient computation, and runs in $\sim 1.17$ seconds. The overall reconstruction process takes roughly 2 hours per scene. This makes the optimization computationally practical while maintaining high reconstruction fidelity and physical consistency. For comparison, 4DGS typically optimizes for $\sim 1$ hour per scene. While DiffWind is slower than 4DGS-style baselines, this is expected because each iteration involves full differentiable physics simulation, tackling a substantially more complex problem. Improving the efficiency of differentiable physical simulators is a broader challenge for the community and a direction for future research, not a limitation unique to our method.

We also evaluated scalability with respect to grid resolution. GPU memory usage during training increases moderately with grid size: $64^3$ uses $\sim 12.3$GB, $128^3$ uses $\sim 13.0$GB, $256^3$ uses $\sim 14.8$GB, and $512^3$ uses $\sim 21.0$GB. This indicates that memory usage remains practical for moderately sized scenes, thanks to Taichi's efficient memory handling. For larger-scale scenarios, multi-GPU parallelization and other scalability strategies can be employed.

## F   MORE DETAILS ON PROMPT DESIGN

We present the prompt template for precise 3D segmentation, as shown in Fig. 13. This template is designed to generate a brief yet accurate segmentation caption when provided with a given image. Next, we design a physical agent based on MLLMs for open-vocabulary semantic reasoning about materials and their physical properties. For each segmented object part, we combine its mask with a canonical reference image and concatenate it with the reference image to form a prompt image. The

garden: A bouquet composed mainly of eucalyptus leaves with sprigs of small pink and white flowers, contained in a beige ceramic pitcher, sits on a dark weathered wooden table. Next to it rests a small vintage-style gramophone accent piece. The camera remains still, no zooming, no closing in, no direction-changing, just remains still. A strong wind gusts from the right, causing the flowers and foliage to sway and dance dramatically and naturally to the left with significant motion.

vase: A vibrant bouquet of white lilies, purple alstroemeria, and green chrysanthemums in a clear glass vase sits on weathered wooden planks. The camera remains still as a strong wind gusts from the right, causing the flowers and foliage to sway and dance dramatically to the left with significant motion.

sock: A large, festive Christmas stocking, primarily red and olive green, featuring white "Merry Christmas" lettering and an attached Santa Claus figurine with a fluffy beard, is clipped by several white clothespins to a dark brown wooden stand. The scene is set indoors against a plain off-white wall and a light-colored tiled floor. The camera remains perfectly static, maintaining a consistent medium shot with no zoom, pan, or tilt. A steady, gentle breeze blows from the right, causing the stocking to sway softly and naturally to the left. The fabric of the stocking should show slight ripples, should exhibit realistic, subtle movements in response to the airflow.

cloth: A light beige jacket hangs from a white plastic hanger. This hanger is hooked onto the side of a small, light-colored wooden shelf-like structure that rests atop a stack of three brown cardboard boxes. The background features dark, plush armchairs and additional cardboard boxes, indicating an indoor setting. The camera maintains a completely static medium shot, with no zoom, pan, or tilt. A strong and steady breeze emanates from the right, causing the jacket to sway violently and naturally towards the left. The fabric of the jacket, especially its sleeves and lower portions, should exhibit realistic ripples and movements due to the wind.

Figure 15: Prompt for video diffusion models.

agent is then guided to identify the object part within the marked area and reason about all possible material types and their associated physical properties. The prompt template is shown in Fig. 17. Note that the MLLM we used here is GPT5. For wind source inference, given the observed RGB video, we first generate the monocular depth video through Video Depth AnythingChen et al. (2025), and then infer the wind source direction using the prompt as illustrated in Fig. 16.

In addition, we utilize the prompt design shown in Fig. 15 to guide video generation models in synthesizing scenes of wind-induced object motion. These synthesized videos are qualitatively compared with our physically-based forward simulation results to evaluate visual realism and motion plausibility, as presented in the main paper.

You are an AI model specialized in analyzing RGB video and depth video.
Focus exclusively on the following task and do not generate unrelated content.
Wait until I provide the video input before producing any result.

**Task Description:**
You will be given a monocular RGB video and its corresponding depth video that
together capture an object being blown by the wind.
Your goal is to estimate the direction of the wind source in 3D space.
⚠️ It is essential that you use **both the RGB video and the depth video together**. Do
not rely only on RGB appearance or 2D motion — you must incorporate depth
information to reconstruct the object's **3D motion trajectory** before inferring
the wind source direction.

**Coordinate System:**
- **x-axis:** horizontal direction (positive to the right)
- **y-axis:** vertical direction (positive upward)
- **z-axis:** perpendicular to the screen, pointing outward toward the viewer
- The origin is at the **bottom-left corner** of the image.

**Instructions:**
1. When the videos are provided, jointly analyze the object's motion using **RGB appearance + Depth structure**.
2. Explicitly use the depth data to lift the motion from 2D image space **into 3D motion space.**
3. Infer the most likely **wind source direction** that causes this motion.
4. Output the result strictly in the following format:
```json
{
  "wind_source_direction": [x, y, z],
  "explanation": "concise explanation showing how both RGB and Depth information
together indicate this direction"
}
```

Figure 16: GPT Prompt for Wind Source Inference.

You are ChatGPT, a large language model trained by OpenAI, based on the GPT-5 architecture.

You need to complete a task related to materials in computer graphics. The task here is object recognition and reasoning about the possible material types and material properties (such as Young's modulus and Poisson's ratio) of the object based on the object information you recognized. I will provide you with a image of a [object_name]. This image is divided into two sub-images. The left one is the original image, and the right one uses blue color to mask the object. I need you to: Identify what object the blue masked area corresponds to. Use your identification results to infer the material type of the object. For the identified material, provide the corresponding Young's modulus and Poisson's ratio. # For instance, I give you an image of a potted plant where the area masked in blue corresponds to the pot. You should identify it as the pot, and then infer the material of the pot, such as ceramic, plastic, or metal, based on the characteristics of the pot. Then, provide the Young's modulus and Poisson's ratio for the inferred material (e.g., ceramic: Young's modulus = 70 GPa, Poisson's ratio = 0.25).

The right sub-image indicates the mask area, and then you could judge based on the left sub-image which is the original image. You must zoom in the corresponding area in the left original image for better recognition. VERY IMPORTANT: For object recognition, you must make your judgment based on this principle: The mask area is all of the object and the area outside the mask does not belong to the object. Let's think step by step for object recognition.

IMPORTANT: Please also combine image information when reasoning about materials, and sort the results according to likelihood. Let's think step by step for material reasoning.

To provide an answer, please provide a short analysis for object recognition and material reasoning. Analysis: - Object recognition: The object in the masked area is [object name], because [reasoning for identification]. - Material reasoning: The possible material types are [material 1], [material 2], ..., because [reasoning for material types]. Material properties: - Young's modulus: [value] GPa (for identified material) - Poisson's ratio: [value] (for identified material) Final answer: [material 1] [material 2] [material 3] ... (sorted by likelihood) Example Output:

Analysis:
- Object recognition: The object in the masked area is the pot, because its shape and appearance match typical pot structures.
- Material reasoning: The possible material types are ceramic, plastic, metal, because pots are commonly made from these materials, based on the visual texture and color.

Material properties:
- Young's modulus: 70 GPa (for ceramic)
- Poisson's ratio: 0.25 (for ceramic)

Final answer: ceramic plastic metal

Figure 17: GPT Prompt for Physical Properties Reasoning.

