# OpenReview forum: "DiffWind: Physics-Informed Differentiable Modeling of Wind-Driven Object Dynamics"
_ICLR.cc/2026/Conference — ICLR 2026 Poster_

### Official Review · Reviewer_8z3H · 2025-10-28

**Soundness:** 3
**Presentation:** 4
**Contribution:** 4
**Rating:** 8
**Confidence:** 2

**Summary:**

This paper introduces **DiffWind**, a physics-informed generative framework that reconstructs and simulates **hidden wind fields** from **multi-view videos** of wind-driven objects.
The core idea is to jointly optimize the **latent wind field** and the **object motion** under physical constraints.
DiffWind represents wind using an **Eulerian grid** and objects using **Lagrangian particles**, coupled via the **Material Point Method (MPM)** for differentiable wind–object interaction.
The method further enforces **fluid dynamics consistency** through a loss derived from the **Lattice Boltzmann Method (LBM)**, ensuring physically plausible flow.

The authors also construct a new dataset (**WD-Objects**) containing both synthetic and real scenes of deformable objects driven by wind.
Experiments show that DiffWind achieves high-quality 3D reconstruction, realistic forward simulation, and plausible “wind relocation” (transferring estimated wind fields to new objects or scenes).

Overall, this is a technically strong paper with detailed derivations — good job by the authors.

**Strengths:**

- **Novel problem formulation:** One of the first attempts to jointly reconstruct *hidden wind fields* and *object dynamics* from visual observations, bridging 3D reconstruction, differentiable physics, and generative modeling.
- **Strong technical depth:** The coupling of the Eulerian grid (wind) and Lagrangian particles (objects) via MPM is elegant and physically grounded. The inclusion of an LBM-based regularization further enforces physical realism.
- **Thorough experiments:** Evaluated on both **synthetic and real** WD-Objects datasets with multiple categories (cloth, flags, plants, etc.), demonstrating reconstruction quality (PSNR, SSIM, LPIPS) and perceptual realism via user studies.
- **Demonstrated versatility:** The model supports *forward simulation* and *wind relocation* tasks, showing potential for cross-scene generalization.

**Weaknesses:**

**W1. Dependence on ideal inputs:** The method requires multi-view, calibrated video and accurate segmentation, which limits its applicability to real-world, in-the-wild scenarios.

**W2. Computational cost:** The framework integrates MPM, LBM, and differentiable rendering, but training time and memory requirements are not reported. Practical efficiency and scalability remain unclear.

**Questions:**

**Q1.** How robust is DiffWind to **incomplete or single-view** input? Could it generalize if only a subset of views is available?

**Q2.** I am curious about the **training data setting**. The quantitative metrics in Table 1 are extremely high — for instance, a PSNR of 52.5 dB suggests a very dense camera setup. How many camera views were used during training?

---

> ### Author Response · Authors · 2025-11-21
> **Response to Reviewer 8z3H(1/2)**
>
> We sincerely thank the reviewer 8z3H for the detailed feedback.  We are encouraged by the recognition of the novelty, technical depth, thorough experiments, and demonstrated versatility of DiffWind. Below, we provide detailed responses to each of the reviewer’s comments.
>
> ### Q1: Dependence on accurate segmentation and multi-view calibrated video.
>
> **Response:**
>
> We sincerely thank the reviewer for raising these points. Foreground segmentation is a **standard preprocessing step** in differentiable physics pipelines, because only the deformable object participates in the simulation and static background regions do not enter the MPM/LBM updates. Modern 2D/3D segmentation techniques (e.g., SAM, EntitySeg and recent 3D-consistent segmentation approaches OmniSeg3D[CVPR2024],  Trace3D[ICCV2025], SAM3D) are already **highly reliable**, making foreground extraction a well-solved preprocessing step.  Our framework fully leverages these segmentation tools in practice, and the segmentation strategies used in our experiments are detailed in the **APPENDIX B**. Moreover, using multi-view calibrated videos is also **common practice** in dynamic reconstruction, especially for fluid-object interaction tasks where single-view cues are insufficient. In our real data capture, camera calibration is performed using Detector-Free SFM[CVPR2024] (without additional calibration billboards), which already provides sufficiently accurate intrinsics and extrinsics for our reconstruction, as described in Sec.4.1. This level of calibration accuracy is straightforward to obtain in practical settings and does not pose difficulties for the experiments conducted in our work.
>
> Based on these considerations, we view the requirements on segmentation and multi-view calibration as practical and commonly adopted in similar pipelines, and they do not pose significant barriers to applying our approach in typical real-world settings. We will clarify these points in the revised paper.
>
> ### Q2: Computational cost and scalability.
>
> **Response:**
>
> All experiments in the paper can be executed on a single NVIDIA RTX 4090 GPU (24GB). Both the LBM and MPM simulators are implemented using the Taichi GPU framework, which provides high-performance GPU execution and allows the entire physics pipeline to run efficiently.
>
> As shown in **Appendix E, Tab.7**, for forward simulation, DiffWind requires less than 1 second per frame to run the complete LBM wind update, MPM object simulation, and differentiable rendering, demonstrating higher efficiency than the existing video generation baselines, for example, VideoCrafter requires ~81.68s to generate 16 frames, DynamiCrafter ~151.47s for 16 frames, and CogVideoX ~292.37s for 50 frames. For reconstruction tasks, each optimization iteration includes the LBM update, MPM simulation, differentiable rendering, and gradient computation, and runs in approximately 1.17 seconds. The overall reconstruction process takes roughly 2 hours per scene. This makes the optimization computationally practical while maintaining high reconstruction fidelity and physical consistency.  For comparison, 4DGS typically optimizes for ~1 hour per scene. While DiffWind is slower than 4DGS-style baselines, this is expected because each iteration involves full differentiable physics simulation, tackling a substantially more complex problem. Improving the efficiency of differentiable physical simulators is a broader challenge for the community and a direction for future research, not a limitation unique to our method.
>
> To evaluate scalability that is closely related to the grid resolution since larger spatial domains require proportionally larger simulation grids, we measured GPU memory usage during training under different grid resolutions: a 64^3 grid uses ~12.3 GB, a 128^3 grid uses ~13.0 GB, a 256^3 grid uses ~14.8 GB, and a 512^3 grid requires ~21.0 GB. This moderate increase indicates that memory usage remains friendly and acceptable for moderately sized scenes, thanks to Taichi’s efficient memory handling. For larger-scale scenarios, multi-GPU parallelization and other scalability strategies could be employed.
>
> We will clarify this point in the revised paper.

---

> ### Author Response · Authors · 2025-11-21
> **Response to Reviewer 8z3H(2/2)**
>
> ### Q3: Robustness to incomplete or single-view input.
>
> **Response:**
>
> We appreciate the reviewer’s question. Since our paper mainly focuses on wind-driven object dynamics modeling, our primary goal is to evaluate the modeling capability under sufficient visual observations. For this reason, we conduct our main experiments using a four-view 360° setup.
>
> However, during the early exploration phase of our project, we also conducted single-view experiments to investigate the limits of reconstruction under extremely sparse observations. These preliminary results, compared against the baseline method GaussianPrediction[SIGGRAPH 2024], show that DiffWind still recovers the visible motion more accurately, though motions on the occluded side cannot be reconstructed reliably due to inherent ambiguity. The novel-view synthesis results(PSNR/SSIM/LPIPS) from this single-view setting are summarized in the table below:
>
> |                    | Ficus             | Alocasia          | ShapeNetPlant     |
> | ------------------ | ----------------- | ----------------- | ----------------- |
> | GaussianPrediction | 16.70/.8220/.1570 | 27.84/.9621/.0379 | 19.19/.9080/.0797 |
> | Ours               | 22.21/.8941/.0455 | 28.86/.9637/.0192 | 21.42/.9480/.0334 |
>
> While the single-view setup naturally yields lower reconstruction accuracy, our results still outperform the baseline, demonstrating that DiffWind maintains strong modeling ability and robustness even under extremely sparse observations.
>
> ### Q4: Why are the PSNR numbers so high? How many camera views were used?
>
> **Response:**
>
> We appreciate the reviewer’s attention to this detail. The high PSNR values in Tab.1 arise from the specific experimental design on our synthetic dataset: We use 4 camera views for dynamic reconstruction, and they provide full 360° coverage of the motion. For synthetic scenes, the camera poses are perfectly accurate, eliminating calibration noise. Before dynamic reconstruction, we first reconstruct the static 3D object, which is the standard pipeline in differentiable physics methods (e.g., PAC-NeRF[ICLR2023], PhysDreamer[ECCV2024]). Because all baselines and our method receive exactly the same high-quality static geometry and view coverage, their PSNR values are also relatively high. Therefore, the reported metrics reflect consistent advantages of DiffWind over baselines, not an unusually dense or privileged capture setup.
>
> We will clarify this point in the revised paper.

---

> > ### Comment · Reviewer_8z3H · 2025-11-27
> >
> > Thank you for your detailed responses. All my concerns have been addressed, and I insist on my positive judgment.
> >
> > As for me, physical interpretability matters. It's a nice try and must require a lot of effort to put fluid mechanics into a differentiable pipeline and finally achieve impressive physical simulation effects using 3D Gaussians. Even though the training efficiency (to generate the physically drivable asset/ to generate video directly) is not as good as video diffusion models currently, 3D consistency and the laws of physics live in the nature of the generated assets, which is extremely difficult for video diffusion models to achieve. The defects of the latter techniques often appear in long-time sequences and rapid viewpoint transformations. Maybe you can defend yourselves from these perspectives. Furthermore, I think the proposed WD-objects dataset is cool. Imagine using such a dataset to pretrain a model that can take a classic Gaussian file as input and output a physically drivable asset. Then the generation time can be largely reduced. wish you good luck.

---

> > > ### Author Response · Authors · 2025-11-27
> > >
> > > We sincerely thank the reviewer for the positive assessment, encouraging remarks, and thoughtful suggestions. We truly appreciate your recognition of our work and your supportive feedback.

---

### Official Review · Reviewer_QPR1 · 2025-10-29

**Soundness:** 2
**Presentation:** 2
**Contribution:** 2
**Rating:** 2
**Confidence:** 4

**Summary:**

This work proposes a method for wind-driven object dynamics reconstruction. It leverages a combination of physics-based simulations and machine learning techniques to accurately model the behavior of objects subjected to wind forces. Given observational data, the method learns to infer the underlying physical parameters used in the Material Point Method (MPM) simulations, enabling realistic reconstructions of object dynamics in windy environments. A physics-aware optimization strategy is employed to improve the fidelity of the reconstructions.

**Strengths:**

1. The coupling of differentiable physics (LBM + MPM) with 3DGS for realistic wind–object interaction modeling is novel.
2. Experiments on synthetic and real-world datasets demonstrate clear performance gains over state-of-the-art methods.
3. The introduction of WD-Objects and the novel “wind retargeting” task broaden research potential.

**Weaknesses:**

1. This work can be viewed as exploration in generative simulation. However, the experimental results in the paper are still toy examples. The author should justify the practicality of the proposed method in real-world applications. What can this method be used for in practice?
2. The novelty of the proposed method should be further emphasized. How does it compare to existing approaches in the literature? Based on PhysGaussian, the authors should discuss more about the differences and improvements over prior optimization-based methods for physical parameter estimation, such as PhysDreamer, Physics3D, DreamPhysics, OmniPhysGS, etc.
3. In the supplementary material, the authors only provide backward reconstruction results for clean-background cases. It seems that the method can only handle simple scenarios, which limits its applicability. The authors should provide more analysis and discussion on the limitations of the proposed method.

**Questions:**

1. What's the computational cost and runtime of the proposed method compared to baselines?
2. How does the method initialize the physical parameters for optimization? A comparison between the initialization and the final learned parameters would be insightful.

---

> ### Author Response · Authors · 2025-11-21
> **Response to Reviewer QPR1(1/3)**
>
> We sincerely thank the reviewer QPR1 for the detailed feedback. Below, we provide **more clarification** and **additional analysis.**
>
> ### Q1: This work can be viewed as exploration in generative simulation.
>
> **Response:**
>
> We clarify that our method is **not** intended as a simple exploration in generative simulation. Our framework is designed for **wind-driven object dynamics modeling**, with a strong focus on the **challenging reconstruction problem**: recovering both the wind force fields and the resulting object dynamics from videos (as pointed out by reviewer 8z3H). This reconstruction task is the central **technical challenge** we address, and most of our framework designs, experiments and evaluations are tailored for it.
>
> Because the modeling framework explicitly integrates LBM for wind-field simulation and MPM for deformable-object dynamics, **it naturally forms a physically coherent simulation system**. As a direct result of this physically grounded formulation, our method inherently supports high-fidelity forward (generative) simulations, including new wind condition simulation and wind-retargeting to new objects.
>
> Our method achieves high-quality dynamic reconstruction of wind-driven object motion, as demonstrated in **Sec.3, Fig.3,4** and **Tab.1,2** of the main paper. The reconstructed wind force fields and object dynamics remain consistent with the observed motion and yield accurate novel-view renderings, significantly outperforming SOTA dynamic reconstruction methods. Furthermore, the forward (generative) simulations produced by our modeling framework exhibit realistic wind-object interactions (**Sec.4.4**, **Fig.5,6** and **Tab.6**) and outperform existing video generation baselines in both visual realism and motion plausibility. These results collectively show that our approach models wind-driven dynamics accurately in reconstruction while also generating physically consistent forward (generative) simulations.
>
> ### Q2: The results are still toy examples. What are the practical uses?
>
> **Response:**
>
> We thank the reviewer for raising this point. Our experiments follow most **prior works in windy object modeling** (e.g., [1] [2] as discussed in Sec.2) and include diverse object types.
>
> Besides the synthetic dataset, as shown in Fig.3, Fig.5, and Video Supp, our experiments exhibit complex wind-object interactions that represent typical real-world scenarios for wind-driven dynamics. The ability to robustly model such complex wind-driven dynamics can support various practical applications, including:
>
> - **VFX** **and animation**: reproducing realistic wind effects for content creation [1, 2].
> - **Digital twins and simulation**: estimating wind loads on flexible structures and natural foliage [3].
> - **Agricultural monitoring**: analyzing wind stress on crops from ordinary footage [4].
> - **AR/VR** **content creation**: generating physically consistent dynamic assets [5].
> - **Robotics and environmental perception**: understanding wind-related object behavior [6].
>
> References:
>
> [1] Runia, T. F. H., Gavrilyuk, K., Snoek, C. G. M., & Smeulders, A. W. M., *Cloth in the Wind: A Case Study of Physical Measurement through Simulation*, CVPR (2020).
>
> [2] Cardona, J. L., Howland, M. F., & Dabiri, J. O., *Seeing the Wind: Visual Wind Speed Prediction with a Coupled Convolutional and* *Recurrent Neural Network*, NIPS (2019).
>
> [3] Mishra, R., Guilmineau, E., Neunaber, I., & Braud, C., *Developing a Digital Twin Framework for Wind Tunnel Testing:* *Validation* *of Turbulent* *Inflow* *and Airfoil Load Applications*, Wind Energy Science, 9, 235–252 (2024).
>
> [4] Pan, Y. et al., *Physics-based algorithm to simulate tree dynamics under wind load*, International Journal of Agricultural and Biological Engineering (IJABE).
>
> [5] Wilson, K., McAdams, A., Leo, H., & Simmons, M., *Simulating Wind Effects on Cloth and Hair in Disney’s Frozen*, SIGGRAPH (2014)
>
> [6] Mittal, M., Guo, K., State, G., Huang, S., et al., *Isaac Lab: A* *GPU**‑Accelerated Simulation Framework for Multi‑Modal Robot Learning*, NVIDIA Research (2025).

---

> ### Author Response · Authors · 2025-11-21
> **Response to Reviewer QPR1(2/3)**
>
> ### Q3: Novelty should be emphasized; comparison to PhysDreamer,etc.
>
> **Response:**
>
> We appreciate the request for further clarification.
>
> Below we summarize the **key novelty** of our work:
>
> (1) **Novel problem formulation.** As highlighted by reviewer 8z3H, we are the first attempts to jointly reconstruct hidden wind fields and object dynamics from visual observations, bridging 3D reconstruction, differentiable physics, and generative modeling.
>
> (2) **Physically grounded modeling.** Our physically principled coupling of Eulerian wind grids and Lagrangian object particles via MPM, with LBM enforcing fluid-consistent wind evolution, makes the formulation significantly more expressive than PhysDreamer, Physics3D, or DreamPhysics, as echoed by reviewer r5yh and reviewer 8z3H.
>
> (3) **Expanded capability over prior optimization-based methods.** Our framework is a unified and generalized end-to-end differentiable framework that integrates LBM, MPM, and 3DGS (reviewer r5yh), which removes limitations of prior work that either modeled only visible dynamics or only simple forces. This goes beyond prior methods that only optimized material parameters or initial velocities under fixed forces.
>
> These points collectively demonstrate that DiffWind introduces substantial conceptual and technical novelty beyond existing optimization-based physical parameter estimation methods such as PhysDreamer, Physics3D, DreamPhysics, and OmniPhysGS.
>
> To further highlight the differences, we summarize a feature-level comparison of our framework and prior methods in the table below:
>
> | Method                              | Ours | PhysGaussian | GSSplashing | VR-GS | PAC-NeRF | PhysDreamer | Physics3D | DreamPhysics | OmniPhysGS |
> | ----------------------------------- | ---- | ------------ | ----------- | ----- | -------- | ----------- | --------- | ------------ | ---------- |
> | Forward Simulation                  | ✔️    | ✔️            | ✔️           | ✔️     | ✔️        | ✔️           | ✔️         | ✔️            | ✔️          |
> | Differentiable-Optimization         | ✔️    | ✘            | ✘           | ✘     | ✔️        | ✔️           | ✔️         | ✔️            | ✔️          |
> | Time-Varying Force Fields           | ✔️    | ✘            | ✘           | ✘     | ✘        | ✘           | ✘         | ✘            | ✘          |
> | Physically-Constrained Optimization | ✔️    | ✘            | ✘           | ✘     | ✘        | ✘           | ✘         | ✘            | ✘          |
> | Dynamic Retargeting                 | ✔️    | ✘            | ✘           | ✘     | ✘        | ✘           | ✘         | ✘            | ✘          |
>
> As shown in **Appendix E.2: Evaluations of Other Baselines**, we already include a comparison to prior optimization-based methods for physical parameter estimation (e.g., PhysDreamer [ECCV2024] and PhysFlow [CVPR2025]), and **they perform poorly on our** **wind-driven** **object dynamics** **reconstruction task**, largely because their optimization pipelines fail to model the coupled wind–object interaction dynamics.
>
> ### Q4: Only clean-background backward reconstruction results are shown, so applicability is limited.
>
> **Response:**
>
> We would like to highlight that background segmentation is a standard preprocessing step in differentiable physics pipelines because the background is static and does not participate in MPM or LBM simulation. This is consistent with **common practice** in differentiable simulation methods (e.g., PAC-NeRF [ICLR2023], PhysDreamer [ECCV2024]), where static regions are routinely masked out before physics-based optimization. Importantly, modern 2D/3D segmentation techniques (e.g., SAM2, EntitySeg, and recent 3D-consistent segmentation approaches OmniSeg3D [CVPR2024], Trace3D [ICCV2025], SAM3D) are already **highly reliable**, making foreground extraction a well-solved preprocessing step. Our framework fully leverages these segmentation tools in practice, and the segmentation strategies used in our experiments are detailed in **APPENDIX B**. Based on these considerations, we therefore believe that the reliance on background segmentation does not materially restrict the applicability of our approach.

---

> ### Author Response · Authors · 2025-11-21
> **Response to Reviewer QPR1(3/3)**
>
> ### Q5: Computational cost and runtime.
>
> **Response:**
>
> We thank the reviewer for raising this question about computation cost, and we provide concrete runtime comparisons below. All timings are measured on a single NVIDIA RTX 4090 GPU, consistent with our experimental setup.
>
> **(1) Reconstruction** **optimization** **runtime.**
>
> For the reconstruction task: 4DGS typically optimizes for ~1 hour per scene, DiffWind requires ~2 hours per scene. For reconstruction tasks, we acknowledge that our method is slower than 4DGS-style baselines, and this difference is expected because each reconstruction time step includes multiple differentiable optimization iterations, involving the full LBM wind update, MPM object simulation, and differentiable rendering. Thus, while DiffWind is slower than 4DGS-style baselines for reconstruction, our method tackles a substantially more complex problem involving full physical simulation, making direct runtime comparison inherently limited in relevance. Improving the efficiency of differentiable physical simulators is a broader challenge for the community and a direction for future research, rather than a problem unique to our method.
>
> **(2)  Forward simulation speed.**
>
> For the forward simulation task: the video-generation baselines such as VideoCrafter, DynamiCrafter, and CogVideoX, VideoCrafter requires ~81.68s to generate 16 frames, DynamiCrafter ~151.47s for 16 frames, and CogVideoX ~292.37s for 50 frames. In contrast, DiffWind performs significantly faster forward simulation: as shown in **Appendix E, Tab.7**, DiffWind requires < 1 second per frame to run the complete LBM wind update, MPM object simulation, and differentiable rendering. Thus, for forward simulation, DiffWind is substantially more efficient than video diffusion models, while providing physically grounded fluid–object interactions that video-generation models cannot capture.
>
> ### Q6: Initialization of physical parameters.
>
> **Response:**
>
> We thank the reviewer for the question. In our framework, the physical parameters consist of two components: material parameters of the object and the wind force fields. Recovering both simultaneously is inherently ambiguous, because different combinations of material stiffness and external forces can produce nearly identical motions. For instance, a small Young’s modulus paired with a small force can generate the same motion as a large Young’s modulus with a large force. To avoid this material–force ambiguity, we first estimate a reasonable material prior using MLLM inference and then fix the material parameters while optimizing the wind force fields. The wind force field is always initialized from zero and fully optimized through differentiable simulation. In controlled synthetic environments with available ground-truth wind force fields, our results **(Appendix E.2, Tab. 9)** show that the reconstructed wind force fields closely match the ground truth, confirming that the optimization can accurately recover physically consistent wind dynamics.
>
> Besides, we also conducted an ablation study to show the robustness of our optimization against material initialization. As shown in **Tab. 4**, even intentionally setting Young’s modulus to very different values (soft, medium, hard) does not significantly affect reconstruction quality.

---

> ### Comment · Reviewer_QPR1 · 2025-11-21
>
> Thanks for the authors’ detailed reply.
>
> I still have concerns regarding parameter initialization. It is possible that the initial parameters are already very close to the optimal solution, which raises doubts about whether the reported improvements truly stem from the proposed method or simply from favorable initialization. It would therefore be helpful if the authors could provide videos or other visualizations showing the dynamics at initialization, as well as at optimization steps 50, 100, 200, and so on, in addition to the final optimized results.
>
> About Q4, I'd like to see visualization results using 3D segmentation techniques on a real-world scene. This could further justify the method's robustness.
>
> If the authors can provide the above results, I'm willing to raise my score.
>
> Also, I have to remind the authors that common practice in prior works doesn't mean the practice is correct or optimal. The author should justify the usage proposed in Q2, instead of just listing them with a simple description. How does the proposed method work in areas such as agriculture?
>
> About the computational time, the whole process requires about 2 hours. However, a video diffusion only takes 5-10 minutes to generate a video, which can be applied to animation and content creation. The author may claim that the video diffusion violates the physics law, but it's highly likely that the generated video has better visual quality compared to Gaussian-Splatting-based techniques.

---

> > ### Author Response · Authors · 2025-11-25
> > **Response to Reviewer QPR1**
> >
> > We sincerely thank the reviewer QPR1 for the follow-up comments and the constructive discussion. We appreciate the reviewer's continued engagement and the thoughtful feedback, which has helped us further clarify and strengthen our work.
> >
> > 1. **About  parameter initialization**: Thank you for the clarification request. As noted in our Q6 response,  to avoid this material–force ambiguity, we first estimate material parameters (e.g., Young' Modulus) using MLLM inference and then fix these material parameters during the optimization. The wind force field is always **initialized from zero** (a standard and non-favorable initialization choice) and fully optimized through differentiable simulation. Despite starting from a zero-force field, our optimization reliably converges to physically consistent wind dynamics and produces high-quality reconstructions. Moreover, as shown in Table 4, we also evaluate our robustness against the initilization of Young’s modulus. We find that even when we intentionally initialize the Young’s modulus with **widely different initial values**, the reconstruction quality remains stable. This demonstrates that our improvements do not stem from favorable initialization, but from the stability and convergence properties of our coupled LBM–MPM optimization framework.
> > 2. **About 3D** **segmentation**: Thank you for the suggestion. In the updated supplementary material **(see seg-3d.pdf)**, we provide new visualizations obtained using OmniSeg3D [CVPR 2024] (as detailed in Appendix B) on several real-world scenes in our experiments. These results demonstrate that modern 3D segmentation techniques provide stable foreground extraction in cluttered and realistic environments.
> > 3. **About prior practice and applications**:
> >
> > - We acknowledge that common practice in prior works does not necessarily imply correctness or optimality. In differentiable physics, the community has been continuously exploring feasible setups, from DiffTaichi [ICLR 2020] to PAC-NeRF [ICLR 2023]. While these examples may look like simple, they reflect the practical constraints required for stable and tractable differentiable simulation, and further development in this direction relies on collective community effort.
> > - As for the application in agriculture, Pan et al. [4] developed a physics-based algorithm to simulate tree dynamics under wind load using a mechanistic model within an L-system framework. Building on this idea, our framework can be used to estimate wind-induced stress and predict plant deformation from ordinary video footage, which may assist basic monitoring tasks in agricultural settings.
> >
> > [4] Pan, Y. et al., *Physics-based algorithm to simulate tree dynamics under wind load*, International Journal of Agricultural and Biological Engineering (IJABE).
> >
> > 4. **Comparison with video diffusion based methods**:
> >
> > - We acknowledge that if the goal is pure animation generation, video diffusion models do not require a reconstruction stage and can directly synthesize a video. However, diffusion-based generation is inherently slow at inference time (e.g., VideoCrafter requires ~81.68s to generate 16 frames, DynamiCrafter ~151.47s for 16 frames, and CogVideoX ~292.37s for 50 frames). In contrast, our method, as well as other physics-based animation approaches, performs a one-time static reconstruction to obtain the 3D model, after which the physics-based animation is fast and reusable. Although the reconstruction introduces additional upfront cost, once completed, our forward simulation runs at **less than one second per frame,** supports rendering from **arbitrary viewpoints** without any further optimization, and imposes no constraints on sequence length. (We also note that the reported two-hour runtime corresponds to dynamic 4D reconstruction. The static reconstruction required for subsequent animation takes about 30 minutes in our experiments.)
> > - As for visual quality, we acknowledge that video diffusion models can often generate visually appealing frames and may avoid certain rendering artifacts associated with 3D Gaussian Splatting. However, diffusion models still lack 3D consistency, stable viewpoint coherence, and physically plausible dynamics. We therefore consider the two directions to be complementary: diffusion excels at appearance realism, while physics-based simulation guarantees dynamic realism, physical controllability, and cross-view consistency. We will make this clearer in the final version as suggested.
> >
> > If you have any further questions, we sincerely welcome continued discussion and are happy to provide additional clarification. We truly appreciate your time and thoughtful evaluation of our work.

---

### Official Review · Reviewer_uA1M · 2025-10-30

**Soundness:** 3
**Presentation:** 4
**Contribution:** 3
**Rating:** 6
**Confidence:** 3

**Summary:**

This paper proposes a physics-informed, differentiable modeling framework designed to simulate and reconstruct wind-driven object dynamics from videos.
It models wind as a grid-based physical field and objects as particle systems, and uses the MPM for object-wind interaction. DiffWind optimizes both wind forces and object dynamics via differentiable rendering and simulation with LBM, ensuring physical consistency.
The objective experimental results are quite impressive.

**Strengths:**

1. The framework allows joint optimization of wind forces and object dynamics and leverages differentiable physics simulation for accurate reconstruction. The use of LBM ensures the wind dynamics adhere to fluid mechanics laws. I think this should be effective and novel.
2. The method outperforms state-of-the-art dynamic scene modeling approaches in both reconstruction accuracy and simulation fidelity.
3. It introduces wind retargeting to enable wind dynamics to be applied to novel objects. This expands the range of its use in simulations and visual effects.
4. It introduces a dataset WD-Objects for modeling wind-driven object dynamics.

**Weaknesses:**

1. As the authors stated, the current implementation focuses on modeling object-level dynamics without accounting for interactions between multiple objects.
2. This method requires accurate segmentation for optimal performance. This may limit its application in less controlled environments and practical scenarios.
3. This paper focuses on continuum objects. What will happen when the method is extended to simulate behaviors in other types of objects?

**Questions:**

1. I wonder whether it becomes computationally intensive for large-scale or complex simulations with LBM and MPM?

---

> ### Author Response · Authors · 2025-11-21
> **Response to Reviewer uA1M**
>
> We sincerely thank the reviewer uA1M for the detailed feedback, the positive evaluation and for highlighting the effectiveness, novelty, and strong empirical performance of our framework. Below, we provide detailed responses to each of the reviewer’s comments.
>
> ### Q1: The method does not account for multi-object interactions.
>
> **Response:**
>
> We agree that multi-object wind–object interaction is an interesting and valuable direction, and as noted in our main paper, this is one of the extensions we plan to explore. The underlying simulation components (MPM for deformable objects and LBM for wind) naturally support multiple interacting bodies, as both solvers inherently handle multi-material and multi-body configurations without requiring changes to their core formulation.
>
> However, we focus on the single-object setting, because robustly modeling object-object collisions in an inverse reconstruction pipeline remains highly challenging. Collision events introduce discontinuous contact dynamics, rapid momentum exchanges, and complex force interactions, all of which significantly complicate gradient-based optimization. Recovering such interactions would require explicit contact inference, collision scheduling, and potentially non-smooth event reasoning, which is beyond the scope of our current framework. Extending the system to reliably handle collision-rich multi-object scenarios is therefore an important avenue for future work.
>
> We will clarify this point in the revised paper.
>
> ### Q2: This method requires accurate segmentation, limiting application in less controlled environments.
>
> **Response:**
>
> We thank the reviewer for raising this thoughtful point. Accurate segmentation is indeed important in differentiable physics pipelines, as static background regions do not participate in the simulation and should not be fed into MPM or LBM. In practice, modern 2D/3D segmentation techniques (e.g., SAM2, EntitySeg and recent 3D-consistent segmentation approaches OmniSeg3D[CVPR2024], Trace3D[ICCV2025], SAM3D) are already **highly reliable**, making foreground extraction a well-solved preprocessing step. Since the simulator only requires the deformable foreground region, foreground extraction is a well-solved preprocessing step rather than a limitation of our method. Our framework fully leverages these segmentation tools in practice, and the segmentation strategies used in our experiments are detailed in **Appendix B**. Therefore, we thus view background segmentation as a practical and standard preprocessing step rather than a limiting factor for our approach.
>
> ### Q3: The paper focuses on continuum objects. What happens to other types of objects?
>
> **Response:**
>
> We thank the reviewer for the helpful question. Our framework is simulator-agnostic and can naturally extend beyond continuum deformables by replacing MPM with an appropriate differentiable dynamics model, such as differentiable spring-mass systems [ECCV 2024] or differentiable FEM solvers like DiffPD [TOG 2022]. The core design, i.e., objects represented as particles and wind represented on a grid, does not depend on any specific simulator. While we use MPM and LBM in our experiments for their robustness and generality, these components can be substituted when modeling different object types.
>
> To be noted, such replacements do not change our overall formulation: the parameterization, optimization objectives, and reconstruction pipeline remain the same.
>
> Thus, extending DiffWind to additional object types primarily requires selecting the appropriate differentiable simulator for the target material or structure, which is part of the future work discussed in our paper.
>
> We will clarify this point in the revised paper.
>
> ### Q4: I wonder whether it becomes computationally intensive for large-scale or complex simulations with LBM and MPM?
>
> **Response:**
>
> We thank the reviewer for the question. As with most differentiable physics solvers, the computational cost increases when scaling to larger spatial domains or more complex flow regimes, because a larger region of wind–object interaction must be discretized.
>
> It is important to note that our framework scales in a predictable and fully GPU-parallel manner: increasing scene size simply increases the grid resolution. This data-parallel structure keeps the method numerically stable and allows it to scale to larger domains as long as sufficient GPU memory is available. For example, on a single NVIDIA RTX 4090, for our reconstruction tasks, a full optimization iteration, which includes the LBM update, the MPM step, and differentiable rendering takes approximately 1.02 seconds for a 64^3 grid, 1.17 seconds for a 128^3 grid, and 2.02 seconds for a 256^3 grid.
>
> These timings illustrate the expected growth in computational cost with increasing grid size. For truly large-scale scenes, one could use multi-GPU parallelization and other scalability techniques, which are beyond the scope of this paper.

---

> > ### Comment · Reviewer_uA1M · 2025-11-25
> >
> > Thank you for your responses. My concerns are mostly addressed. I will keep my positive score.

---

> ### Comment · Reviewer_uA1M · 2025-11-26
> **Another question about subjective comparison**
>
> Upon revisiting the subjective video comparisons in the supplementary material, I noticed that CogVideoX and the other video models appear to have very low frame rates. Could you provide the exact frame rate, and clarify whether it is possible to compare against more advanced video diffusion models? At the very least, there should be a subjective comparison with open-source models such as HunyuanVideo and Wan.

---

> > ### Author Response · Authors · 2025-11-27
> > **Response to Reviewer uA1M**
> >
> > We sincerely thank the reviewer for the careful examination and insightful comments regarding the subjective video comparisons in the video supplementary material. We appreciate the opportunity to clarify the frame rate of the generated videos and the limitations in comparing different video generation models.
> >
> > As noted, the low frame rates observed in CogVideoX and other video generation models are primarily due to the **maximum generated frame numbers constraint** imposed by each model. This limitation restricts the frame number of the generated videos, resulting in lower frame rates when compared side-by-side. To provide transparency, we have compiled a summary table below that lists **Frame Numbers, Video Duration, and Maximum Generated Frames** for each video generation model used in our comparisons, where the Video Duration is determined by each model’s default settings:
> >
> > | Method        | Frame Numbers | Video Duration | Maximum Generated Frames |
> > | ------------- | ------------- | -------------- | ------------------------ |
> > | Cogvideox     | 49            | 6s             | 49                       |
> > | SVD           | 25            | 4s             | 25                       |
> > | VideoCrafter  | 16            | 1.6s           | 16                       |
> > | DynamiCrafter | 16            | 2s             | 16                       |
> > | Wan           | 81            | 5s             | 81                       |
> > | Hunyuan1.5    | 129           | 5s             | 129                      |
> >
> > As suggested, we have further included subjective comparisons with open-source models such as **HunyuanVideo** and **Wan** in the uploaded supplementary material(**see Hunyuan_Wan.mp4**). These additional results confirm that even these more advanced video diffusion models still struggle to generate physically plausible motions, consistent with our original observations.
> >
> > We hope this clarifies the frame rate situation and highlights the current challenges faced by video generation models in producing physically coherent sequences.

---

> > > ### Comment · Reviewer_uA1M · 2025-11-27
> > >
> > > I am satisfied with the authors’ response. My concerns are addressed.

---

### Official Review · Reviewer_r5yh · 2025-10-30

**Soundness:** 4
**Presentation:** 4
**Contribution:** 4
**Rating:** 8
**Confidence:** 3

**Summary:**

The paper proposes DiffWind, a physics-informed, differentiable framework that jointly reconstructs an invisible, time-varying wind field and the visible, deformable object motion from sparse-view RGB videos. The authors also introduce WD-Objects (synthetic + real scenes) and report improved novel-view rendering and physically plausible simulations, including wind retargeting and forward simulation under new wind conditions.

**Strengths:**

1. The motivation is quite clear. The paper is well written and easy to follow.
2. The creative combination of LBM + MPM + 3DGS removes limitations of prior work that either modeled only visible dynamics or only simple forces.
3. The wind retargeting demonstrates a strong capability to generalize the wind to other objects.
4. On both synthetic and real data, DiffWind outperforms state-of-the-art dynamic 3DGS baselines on novel view synthesis.

**Weaknesses:**

1. The method explicitly optimizes only the wind force field while keeping material parameters fixed after MLLM “physical agent” reasoning; this makes it hard to disentangle whether observed motion comes from wind magnitude or material stiffness/damping.
2. Evaluation on real data relies on image metrics (PSNR/SSIM/LPIPS) and a user study, but no direct wind-field measurements are reported. Given that there are no public datasets for wind-driven dynamics, it would strengthen claims to instrument selected scenes with anemometers or PIV.

**Questions:**

1. What are the failure modes when the wind is highly turbulent or when Reynolds numbers push LBM discretization limits at the chosen grid resolution?
2. How does performance scale with grid size (e.g., $128^3$) vs. runtime, and is there an adaptive meshing strategy planned?

---

> ### Author Response · Authors · 2025-11-21
> **Response to Reviewer r5yh(1/2)**
>
> We sincerely thank the reviewer r5yh for the detailed feedback. We are encouraged by the recognition of our motivation, technical design, and experimental results. Below, we provide detailed responses to each of the reviewer’s comments.
>
> ### Q1: Material parameters are fixed after MLLM, making wind/material effects hard to disentangle.
>
> **Response:**
>
> We thank the reviewer for the insightful question. Recovering both material parameters and wind force fields simultaneously is inherently ambiguous, because different combinations of material stiffness and external forces can produce nearly identical motions. For instance, a small Young’s modulus paired with a small force can generate the same motion as a large Young’s modulus with a large force.
>
> For this reason, we use MLLM-based reasoning to obtain **reasonable material priors** (Sec. 3.2, Appendix F) and keep these parameters fixed during optimization. This design choice intentionally removes the material–force coupling ambiguity and allows the optimizer to concentrate on estimating the wind force fields, which are the primary factors governing the observed dynamics in our setting.
>
> We will **clarify this point in the revised paper**.
>
> ### Q2: No direct wind-field instrumentation in real scenes.
>
> **Response:**
>
> We appreciate this suggestion. As the reviewer notes, there are currently no publicly available datasets with ground-truth wind fields. Instrumenting real scenes with anemometers or particle-image velocimetry (PIV) would indeed provide valuable validation. However, such instrumentation requires controlled setups, multi-sensor calibration, and specialized hardware that goes beyond what can be feasibly collected for in-the-wild videos.
>
> For this reason, our real-scene evaluation focuses on novel-view rendering quality (**Tab. 1,2**), and a user study (**Tab. 6**). Importantly, in controlled synthetic environments where ground-truth wind fields are available, we already provide a quantitative comparison in the **Appendix E.2, Tab.9.** These results show that the reconstructed wind force fields closely match the ground-truth fields, demonstrating that the proposed optimization can accurately recover physically consistent wind dynamics.
>
> We agree that incorporating real-world wind-field instrumentation would be a valuable extension, and we plan to explore such data collection for future work.

---

> ### Author Response · Authors · 2025-11-21
> **Response to Reviewer r5yh(2/2)**
>
> ### Q3: Failure modes under highly turbulent wind or large Reynolds numbers.  Response:
>
> We thank the reviewer for raising this insightful question. It touches on an important aspect of fluid simulation and reflects a deep understanding of the underlying numerical modeling.
>
> All numerical solvers for fluid dynamics exhibit limitations when the flow becomes highly turbulent or when the local Reynolds number exceeds the stability range of the scheme. We adopt LBM due to its relative numerical stability and efficiency. In regimes with strong turbulence, any solver, including LBM, may struggle to fully resolve small scale vortical structures or highly chaotic flow patterns. These limitations arise from the intrinsic properties of numerical solvers and are not specific to our method.
>
> In our reconstruction framework, however, LBM is used **as a physics-informed constraint**. This means its role is to regularize the wind reconstruction toward physically plausible behaviors. As a result, the optimization can still handle moderately complex and unsteady wind fields, but consistent with the limitations of the underlying discretization, it will not capture the full spectrum of fine-grained turbulent structures.
>
> We will clarify this point in the revised paper.
>
> ### Q4: How does performance scale with grid size and adaptive meshing strategy planned?
>
> **Response:**
>
> We thank the reviewer for this insightful question. To better understand the impact of grid resolution on performance and reconstruction quality, we conducted a **new ablation study** comparing grid sizes of 64^3, 128^3, and 256^3. The reconstruction results(PSNR/SSIM/LPIPS) for each setting are summarized below:
>
> |       | PSNR  | SSIM  | LPIPS $\times10^{-1}$ |
> | ----- | ----- | ----- | --------------------- |
> | 64^3  | 49.08 | .9991 | .0046                 |
> | 128^3 | 51.31 | .9995 | .0030                 |
> | 256^3 | 51.90 | .9997 | .0024                 |
>
> These results indicate that the overall reconstruction quality remains stable.
>
> **Runtime scaling.**
>
> As reported in the main paper, all experiments are executed on a single NVIDIA RTX 4090 GPU. Under the default 128^3 grid resolution, each optimization iteration, which includes the LBM update, the MPM simulation, and the differentiable rendering, takes ~1.17 seconds. For reference, iteration on a 64^3 grid takes ~1.02 seconds, while a 256^3 grid requires ~2.02 seconds.
>
> These measurements indicate that 128^3 achieves a good balance between reconstruction accuracy and runtime. The 256^3 grid yields only slight quality gains but noticeably increases computational cost, while 64^3 is faster but reduces reconstruction fidelity.
>
> As for adaptive meshing strategy, the Taichi GPU framework already provides an adaptive meshing strategy via hierarchical sparse grids, allocating spatial resolution where needed and efficiently handling regions with high activity.
>
> We will update the new ablation study in the revised paper.

---

> > ### Comment · Reviewer_r5yh · 2025-11-25
> >
> > Thanks for your thorough responses. I am satisfied with them and have no further questions.

---

### Author Response · Authors · 2025-12-02
**Revision Summary**

*Dear AC and Reviewers,*

We are greatly encouraged by the supportive assessments of ***Reviewers 8z3H, uA1M and r5yh*** on the value of our work. And we sincerely appreciate all reviewers for their suggestions and have incorporated their feedback in the revised manuscript. All updates in the revised manuscript are highlighted for clarity.

**Main Paper & Appendix**

- We have clarified in Section 3.3 that we fix material parameters after MLLM initialization to avoid the inherent ambiguity between material properties and wind force fields. (***Reviewer r5yh, W1; Reviewer QPR1, Q2***)
- We have added the ablation study of different grid resolutions to the main paper (Section 4.3, Table 6) to address how performance scales with grid size. (***Reviewer r5yh, Q2***)
- We have expanded the main paper (Section 5) to clarify the technical challenges of handling multi-object interactions and to explain how our framework extends to non-continuum object types via appropriate differentiable physical simulators. (***Reviewer uA1M, W1&W3***)
- We have added a discussion about highly turbulent wind conditions to the Appendix A.2. (***Reviewer r5yh, Q1***)
- We have added computational cost and scalability to the Appendix E.3, providing detailed runtime, memory usage, and GPU requirements for forward simulation and reconstruction. (**Reviewer uA1M, Q1; Reviewer QPR1, Q1; Reviewer 8z3H, W2**)

**Supplementary Materials**

- We include an original version of our paper in the supplementary materials.
- As suggested by ***Reviewer QPR1***, we provide visualizations of 3D segmentation results from our experiments (see **seg-3d.pdf**). These results further validate the reliability of modern 3D segmentation methods, and we consider 3D segmentation a robust preprocessing step that does not pose significant barriers to applying our approach in typical real-world settings. (***Reviewer uA1M, W2; Reviewer QPR1, W3; Reviewer 8z3H, W1***)
- As suggested by ***Reviewer uA1M***, we have further included subjective comparisons with more advanced video diffusion models such as **HunyuanVideo** and **Wan** (**see Hunyuan_Wan.mp4**). These additional results confirm that even these more advanced video diffusion models still struggle to generate physically plausible motions, consistent with our original observations.

We sincerely appreciate the reviewers’ time and constructive feedback, which have substantially improved the clarity and quality of our manuscript. We also sincerely thank the AC for their extra work in managing the review process under difficult circumstances this year.

*Best,*

*Authors*

---

### Meta-Review · Area_Chair_97RV · 2026-01-14

**Summary:**

This paper focuses on recovering both invisible wind force fields and deformable objects from windy videos, a problem that is non-trivial for existing generative models. Most reviewers acknowledge the paper’s strong motivation and novelty. The primary concerns raised relate to the method’s reliance on idealized inputs, its computational cost, and its practical applicability. The authors’ rebuttal effectively addresses most of these concerns and is positively received by three reviewers. While one reviewer continues to question the practical usage of the proposed method, these issues extend beyond the scope of the paper and, to some extent, the current state of the field. Based on these considerations, I believe the paper meets the acceptance bar for ICLR.

**Reviewer Concerns:**

There are four reviewers: r5yh, uA1M, QPR1, and 8z3H. Among them, r5yh, uA1M, and 8z3H explicitly state that their concerns have been well addressed and indicate that they will maintain their positive scores. Based on the additional results and demonstrations, I agree that their concerns have been adequately resolved. Reviewer QPR1, while most of his/her concerns have been addressed, continues to express reservations regarding the practical applicability of the proposed method.

**Reviewer Scores:**

1. r5yh: The reviewer acknowledges the authors’ rebuttal and raises no further questions, and is expected to maintain the positive score.
2. uA1M: The reviewer explicitly states that the positive score will be maintained.
3. QPR1: As the reviewer continues to raise questions regarding practical applicability after the discussion, the score is unlikely to change.
4. 8z3H: The reviewer clearly indicates that he/she will stand by the positive score.

---

### Decision · Program_Chairs · 2026-01-26

Accept (Poster)